# Non-invasive assessment of normal and impaired iron homeostasis in the brain

Shir Filo ®[1] ✉, Rona Shaharabani ®[1], Daniel Bar Hanin[1], Miriam Adam[1], Eliel Ben-David ®[2], Hanan Schoffman[3], Nevo Margalit[4], Naomi Habib ®[1], Tal Shahar[3,4,5,6] & Aviv A. Mezer ®[1,6]

Strict iron regulation is essential for normal brain function. The iron homeostasis, determined by the milieu of available iron compounds, is impaired in aging, neurodegenerative diseases and cancer. However, non-invasive assessment of different molecular iron environments implicating brain tissue's iron homeostasis remains a challenge. We present a magnetic resonance imaging (MRI) technology sensitive to the iron homeostasis of the living brain (the $r_1$-$r_2^*$ relaxivity). In vitro, our MRI approach reveals the distinct paramagnetic properties of ferritin, transferrin and ferrous iron ions. In the in vivo human brain, we validate our approach against ex vivo iron compounds quantification and gene expression. Our approach varies with the iron mobilization capacity across brain regions and in aging. It reveals brain tumors' iron homeostasis, and enhances the distinction between tumor tissue and non-pathological tissue without contrast agents. Therefore, our approach may allow for non-invasive research and diagnosis of iron homeostasis in living human brains.

Iron is the most abundant trace element in the human body[1]. It participates in fundamental processes such as oxygen transport, cellular metabolism, myelin formation, and the synthesis of neurotransmitters[1–4]. Therefore, strict iron regulation is essential for maintaining normal brain function. Brain tissue's iron homeostasis could be characterized by the tissue's molecular iron environment, determined by the specific milieu of available iron compounds, their iron binding capacities and aggregation states. Importantly, the molecular iron environment varies between cell types and across brain regions[3,5–7].

Disrupted iron homeostasis plays a major role in normal aging and in neurodegenerative diseases such as Parkinson's disease (PD), Alzheimer's disease (AD), multiple sclerosis, Friedreich's ataxia, aceruloplasminaemia, neuroferritinopathy, Huntington's disease, and restless legs syndrome[1,2,5–10]. The two iron compounds most involved in iron homeostasis are transferrin and ferritin[3]. Transferrin, the main iron transport protein, carries iron from the blood into brain tissue, while ferritin, the main iron storage protein, stores excess iron atoms. When iron concentrations exceed the capacity of iron-binding proteins, oxidative stress, and cellular damage can occur[10]. For example, the ratio of transferrin to iron, which reflects iron mobilization capacity, differs between the brains of elderly controls and patients (AD and PD)[7]. In addition, reduction in neuromelanin-iron complexes in the substantia nigra and the locus coeruleus is considered a biomarker for PD and AD[11,12].

Impaired homeostasis of the molecular iron environment also have been reported in cancer cells[13,14]. Tumor cell proliferation requires a modulated expression of proteins involved in iron uptake. In addition, iron may affect the immune surveillance of tumors[15].

[1]The Edmond and Lily Safra Center for Brain Sciences, The Hebrew University of Jerusalem, Jerusalem, Israel. [2]The Department of Radiology, Shaare Zedek Medical Center, Faculty of Medicine, The Hebrew University of Jerusalem, Jerusalem, Israel. [3]The Laboratory of Molecular Neuro-Oncology, Shaare Zedek Medical Center, Faculty of Medicine, The Hebrew University of Jerusalem, Jerusalem, Israel. [4]The Department of Neurosurgery, Shaare Zedek Medical Center, Faculty of Medicine, The Hebrew University of Jerusalem, Jerusalem, Israel. [5]The Department of Neurosurgery, Tel Aviv Sourasky Medical Center, Affiliated with Sackler Faculty of Medicine, Tel Aviv University, Tel Aviv, Israel. [6]These authors contributed equally: Tal Shahar, Aviv A. Mezer. ✉e-mail: shir.filo@mail.huji.ac.il

Therefore, the availability of iron in the tumor cells' microenvironment may affect their survival and growth rate, and subsequently the course of the disease. For example, meningioma brain tumors[16], compared to non-pathological tissue, were shown to contain a higher concentration of ferrimagnetic particles and abnormal expression of iron-related genes[17,18]. These findings suggest there are detectable differences in iron homeostasis between brain tumors and normal brain tissue.

The extensive implications of impaired iron homeostasis in normal aging, neurodegeneration and carcinogenesis suggest that assessment of iron homeostasis in the living brain would be highly valuable for diagnosis, therapeutic monitoring, and understanding pathogenesis of diseases[4]. Iron's paramagnetic properties make magnetic resonance imaging (MRI) a perfect candidate for non-invasive estimation of iron content in brain tissue. In particular, iron is a major contributor to the longitudinal and effective transverse relaxation rates, $R_1$ and $R_2^*$ respectively[19–22]. These relaxation rates can be measured using quantitative MRI (qMRI) techniques[23–25]. Indeed, in vivo studies often use these qMRI measurements as a proxy for iron presence in brain tissue[21,26–30]. However, a major limitation of current MRI techniques is that they lack information regarding the state of iron homeostasis, as they do not have the sensitivity to discriminate between different molecular environments of iron in the brain[4].

Early in vitro and postmortem works suggest that different iron environments can be distinguished by their iron relaxivity[30–34]. The iron relaxivity is defined as the dependency of MR relaxation rates on the iron concentration[35]. It was shown that iron relaxivity varies with the specific environment in which the iron resides[30–34]. However, this approach requires direct estimation of the tissue iron concentration, which can only be acquired in vitro or postmortem. Due to this major limitation, until now the phenomenon of iron relaxivity could not be studied in living humans.

Here we propose an in vivo iron relaxivity approach sensitive for the state of iron homeostasis in the brain. Our approach fully relies on MRI parameters, and does not require estimation of the tissue iron concentration, thereby allowing non-invasive assessment of different molecular iron environments in the living brain (Fig. 1). We exploit the distinct iron relaxivities of the MR relaxation rates, $R_1$ and $R_2^*$, to construct a biophysical model of their linear interdependency, which we label the $r_1$-$r_2^*$ relaxivity. Using the $r_1$-$r_2^*$ relaxivity, we argue that the distinct iron relaxivity of different molecular iron environments can be estimated in vivo. We confirm this hypothesis based on a unique validation framework. First, we used a bottom-up strategy in which we evaluated the $r_1$-$r_2^*$ relaxivity of different iron environments in vitro. Next, we used a top-down strategy in which we measured the $r_1$-$r_2^*$

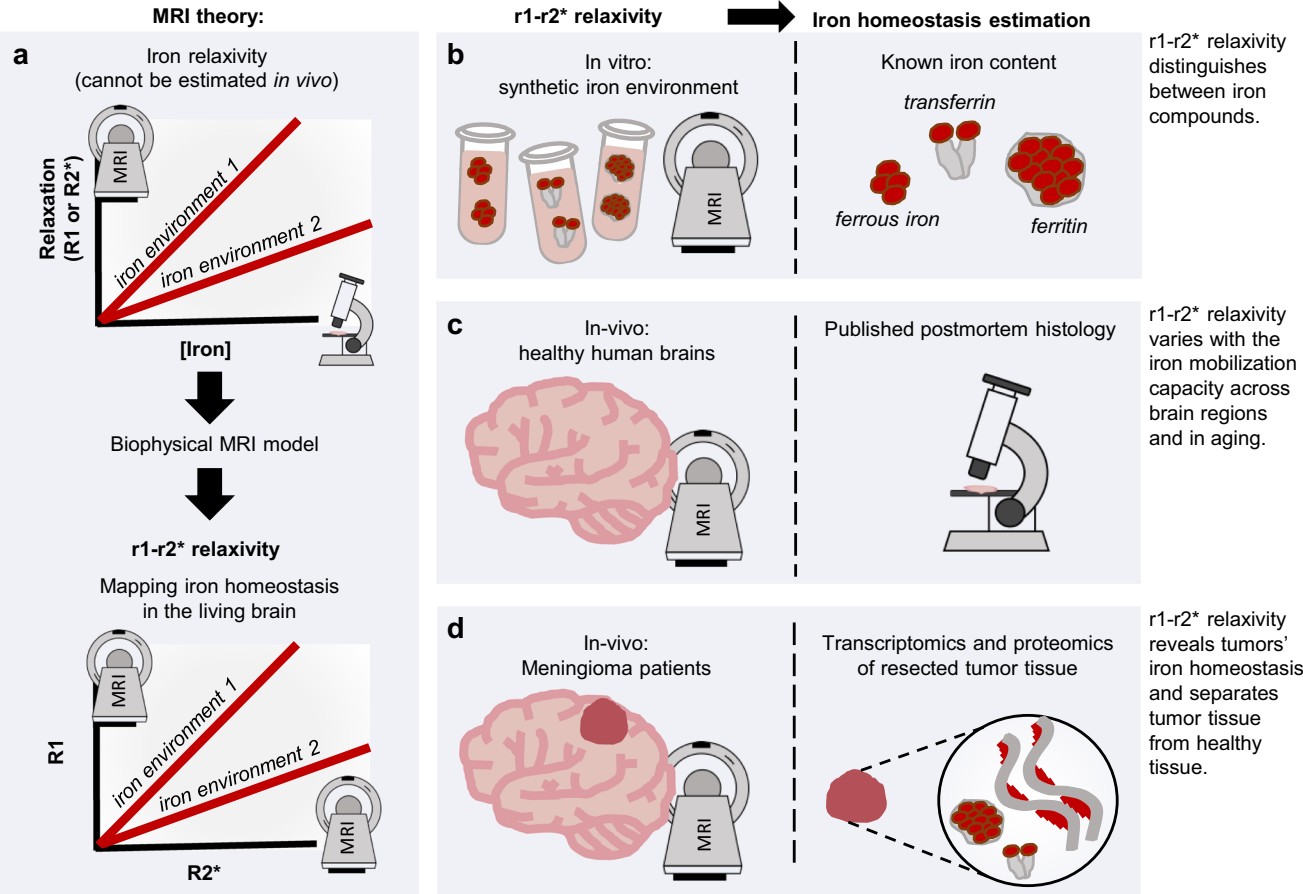

**Fig. 1 | Non-invasive assessment of normal and impaired iron homeostasis in the brain. a** MRI theory: Early in vitro and postmortem works suggest that different iron environments can be distinguished by their iron relaxivity, i.e. the dependency of MR relaxation rates on the iron concentration. This approach requires direct estimation of the tissue iron concentration, which can only be acquired in vitro or postmortem. Here we propose an in vivo iron relaxivity approach. We exploit the distinct iron relaxivities of the MR relaxation rates, R1 and R2*, to construct a biophysical model of their linear interdependency, which we label the $r_1$-$r_2^*$ relaxivity. Using the $r_1$-$r_2^*$ relaxivity, we argue that the state of iron homeostasis in the brain can be estimated in vivo. **b** In vitro, the $r_1$-$r_2^*$ relaxivity distinguishes between different synthetic iron environments (ferritin, transferrin and ferrous iron ions). **c** Comparison between in vivo MRI scans of the healthy human brain and published postmortem histology. The $r_1$-$r_2^*$ relaxivity varies with the iron mobilization capacity across brain regions and in aging. **d** Direct comparison between in vivo MRI scans of meningioma patients and transcriptomics and proteomics of the resected tumor tissues. The $r_1$-$r_2^*$ relaxivity reveals tumors' iron homeostasis and separates tumor tissue from healthy tissue.

relaxivity in human brains in vivo and compared it to ex vivo quantification of iron compounds and gene expression, both at the group and the single-subject levels. In healthy subjects, we assessed the biological correlates of the $r_1$-$r_2^*$ relaxivity, and compared it to other MR contrasts. In meningioma patients, we tested the $r_1$-$r_2^*$ relaxivity contrast between pathological and non-pathological tissues, and compared MRI measurements to ex vivo iron homeostasis estimates of tumors. Therefore, we provide a well-validated MRI framework with promising implications for the non-invasive research and diagnosis of normal and impaired iron homeostasis in living human brains.

## Results

### The theoretical basis for the $r_1$-$r_2^*$ relaxivity

The iron relaxivity is defined based on the linear relationship between the relaxation rates ($R_1$ and $R_2^*$) and the iron concentration ([$IC$])[35]:

$$R_1 = r_{(1,IC)}[IC] + c_1 \qquad R_2^* = r_{(2,IC)}[IC] + c_2 \qquad (1)$$

The slopes of these linear dependencies, $r_{(1,IC)}$ and $r_{(2,IC)}$, represent the iron relaxivities of $R_1$ and $R_2^*$, which were shown to have different values for different iron environments[30–33]. $c_1$ and $c_2$ are constants. Notably, the iron relaxivities require estimation of the iron concentration ([$IC$]), thereby limiting this approach to in vitro and ex vivo studies.

Here we propose a theory which advances the relaxivity model and provides in vivo iron relaxivity measurements for identifying different iron environments in the brain. We take advantage of the fact that $R_1$ and $R_2^*$ are governed by different molecular and mesoscopic mechanisms[36–38], and therefore each of them may have a distinct iron relaxivity in the presence of paramagnetic substances. Based on our theoretical framework ("In vivo iron relaxivity model" in Methods), the linear dependency of $R_1$ on $R_2^*$ can be described by the following equation:

$$R_1 = \frac{r_{(1,IC)}}{r_{(2,IC)}} * R_2^* + c \qquad (2)$$

The slope of this linear dependency ($\frac{r_{(1,IC)}}{r_{(2,IC)}}$) is defined as the $r_1$-$r_2^*$ relaxivity. It represents the ratio of the iron relaxivities, $r_1$ and $r_2^*$, which are sensitive to the molecular environment of iron. Therefore, we hypothesize the $r_1$-$r_2^*$ relaxivity reveals the distinct properties of different molecular iron environments. $c$ is the intercept (residual $R_1$ not explained by $R_2^*$). Notably, in the brain, the $r_1$-$r_2^*$ relaxivity could be affected by the entire milieu of available iron compounds, their iron binding capacities and aggregation states (i.e. the molecular iron environment). For an extension of the relaxivity model in the presence of a heterogeneous iron environment and myelin see Supplementary Section 1.

### In vitro validation for the sensitivity of the iron relaxivity to molecular iron environments

Before implementing this approach in the living human brain, we validated our theory by manufacturing in vitro samples of different iron compounds in a synthetic cellular membrane environment. These samples were scanned in the MRI to verify that different iron environments have different iron relaxivities. We then tested whether our $r_1$-$r_2^*$ relaxivity theory could reveal these different relaxivities.

We prepared samples of transferrin, ferritin and ferrous iron ions in different cellular-like environments (water, liposomal, or protein environments; To achieve physiological iron concentrations the transferrin concentrations are higher than the ones measured in vivo[3,4]). These highly controlled synthetic iron environments were scanned for $R_1$ and $R_2^*$ mapping. We found that both $R_1$ and $R_2^*$ increased with the concentration of iron compounds (Fig. 2a, b). The

rate of this increase, defined as the iron relaxivity, was different for different iron environments (Fig. 2c, p(one-sided ANCOVA) = 1.5 × $10^{-74}$, F-statistics(5) = 475 for R1, p(one-sided ANCOVA) = 2.6 × $10^{-73}$, F-statistics(5) = 450 for R2*). We show that $R_1$ and $R_2^*$ change both with the type and concentration of iron, thereby making it impossible to distinguish between iron environments with these measurements. For example, $R_1$ increased with the ferritin concentration, but also was higher for ferritin compared to transferrin (Fig. 2a, b, Supplementary Fig. 1a, b). Consequently, similar $R_1$ values can be obtained for ferritin and transferrin, depending on their concentrations (Supplementary Fig. 1b). This ambiguity can be resolved by the iron relaxivity, which differentiated the iron environments, and was consistent when computed over samples with higher or lower concentrations (Supplementary Fig. 1c). Therefore, we find that the iron relaxivity changes with the molecular environment of iron and is independent of the iron concentration.

Ferritin binds thousands iron ions more than transferrin[3]. Thus, we wanted to exclude the possibility that the relaxivity differences were being driven by the different iron ion concentration. We estimated the iron ion concentrations for the different molecular iron environments, and verified that ferritin, transferrin and ferrous iron ions indeed have different iron relaxivities even when accounting for the discrepancies in iron concentrations (Supplementary Section 2).

### The $r_1$-$r_2^*$ relaxivity reveals the distinct iron relaxivities of different iron environments

In agreement with previous findings[30–33], our in vitro experiments indicate that iron relaxivity is sensitive to different iron environments. The iron relaxivity represents the dependency of relaxation on the iron concentration, which cannot be estimated in vivo. The $r_1$-$r_2^*$ relaxivity is defined as the dependency of $R_1$ on $R_2^*$, and thus only relies on MRI measurements that can be estimated in vivo. Based on our theory, we argue that two iron environments with different iron relaxivities are also likely to differ in their $r_1$-$r_2^*$ relaxivities. We validated this hypothesis using synthetic iron-containing samples. As predicted by our theoretical model, iron environments with different iron-relaxivities had different $r_1$-$r_2^*$ relaxivities (Fig. 2d, e, p(one-sided ANCOVA) = 1.2 × $10^{-40}$, F-statistics(5) = 103). Notably, as suggested by our theoretical formulation, the $r_1$-$r_2^*$ relaxivity provides a good MRI approximation for the ratio between the iron relaxivities of $R_1$ and $R_2^*$ ($\frac{r_{(1,IC)}}{r_{(2,IC)}}$, see Eq. (2), Fig. 2e). For liposomal transferrin and ferrous iron ions, the model's prediction is slightly outside the confidence interval of the experimental results. This seems to be related to $R_2^*$ estimations, as the prediction improves when replacing $R_2^*$ with $R_2$ (Supplementary Section 3). Similar to the iron relaxivities, the $r_1$-$r_2^*$ relaxivity is consistent across iron concentrations (Supplementary Fig. 1d). Hence, the $r_1$-$r_2^*$ relaxivity is more sensitive to the molecular iron environment than $R_1$ and $R_2^*$ by themselves. In addition, we validated that the $r_1$-$r_2^*$ relaxivity is sensitive to the paramagnetic properties of iron-binding proteins. We found that apo-transferrin (transferrin which is not bound to iron) has a much smaller $r_1$-$r_2^*$ relaxivity compared to iron-bound transferrin (p(one-sided ANCOVA) = 5.6 × $10^{-8}$, F-statistics(1) = 44.38; Supplementary Fig. 5). This implies that it is paramagnetic properties that induce the $r_1$-$r_2^*$ relaxivity that we measure. Taken together, these results validate our theory, indicating that the $r_1$-$r_2^*$ relaxivity can be used to measure iron relaxivity in vivo for exposing the distinct paramagnetic properties of different molecular iron environments.

Brain tissue includes a complex milieu of iron compounds. Therefore, we tested in vitro the $r_1$-$r_2^*$ relaxivity of a heterogenous molecular iron environment of ferritin-transferrin liposomal mixtures (Supplementary Section 4). We found that changing the transferrin-ferritin ratio leads to considerable changes in the $r_1$-$r_2^*$ relaxivity, even in mixtures with low ratio of transferrin compared to ferritin as in the

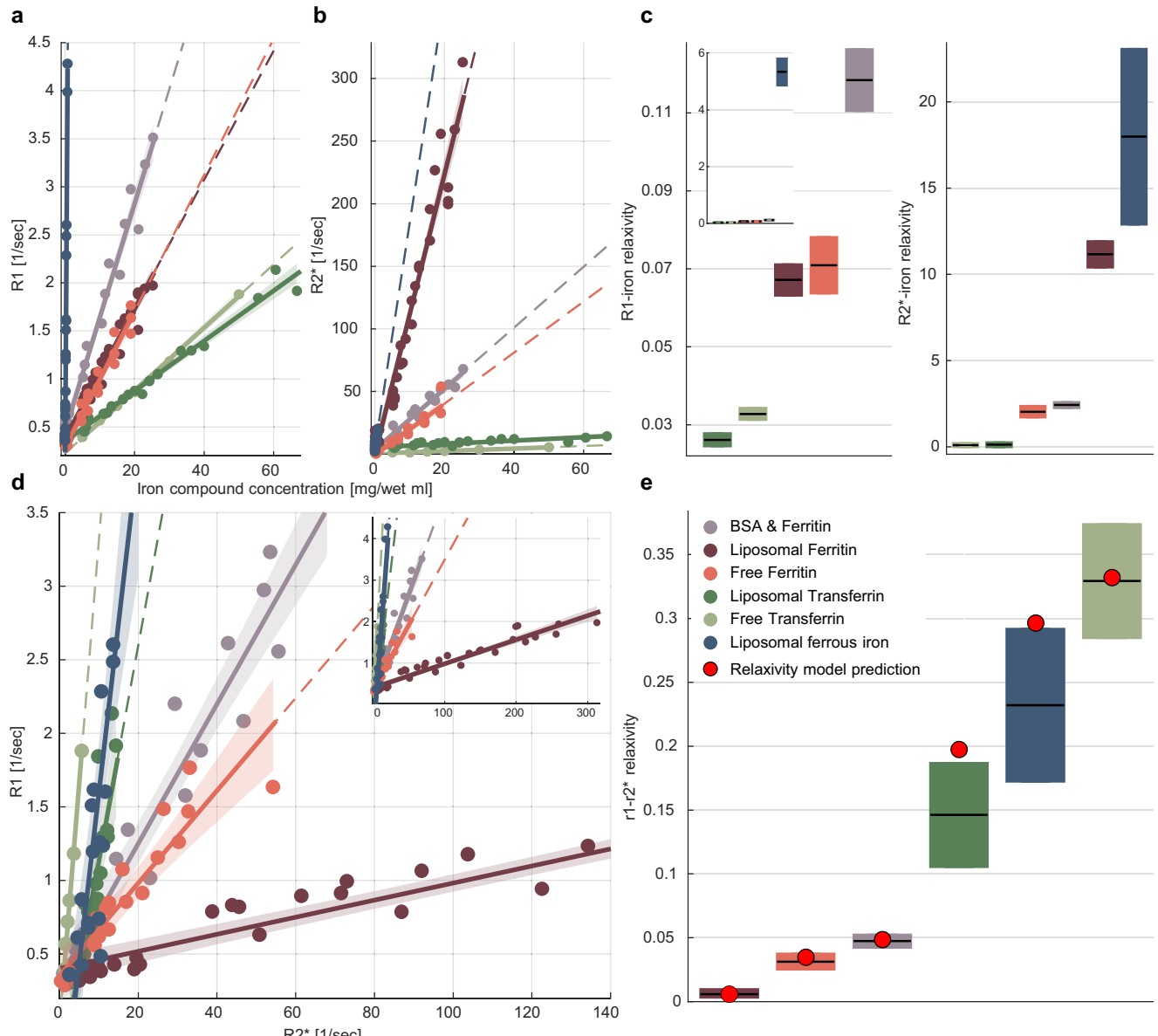

**Fig. 2 | In vitro validation of the non-invasive framework for assessing the iron environments. a**, **b** The dependency of R1 and R2* on the iron compound concentrations for samples of different iron environments: free ferritin ($N = 20$), liposomal ferritin ($N = 36$), bovine serum albumin (BSA)-ferritin mixture ($N = 22$), free transferrin ($N = 6$), liposomal transferrin ($N = 22$) and liposomal ferrous iron ions ($N = 20$). Data points represent median values of biologically independent samples with varying concentrations relative to the water fraction ([mg/wet ml]). The linear relationships between relaxation rates and iron compounds concentrations are marked by solid lines. We define the slopes of these lines as the iron relaxivities. Dashed lines represent extrapolation of the linear fit. Shaded areas represent the 95% confidence bounds. **c** The iron relaxivity of R1 and R2* is different for different iron environments (p(one-sided ANCOVA) = $1.5 \times 10^{-74}$ and $2.6 \times 10^{-73}$ respectively). Iron relaxivities are calculated by taking the slopes of the linear relationships shown in (**a**, **b**), and are measured in [sec-1/(mg/wet ml)]. For each box, the central mark is the iron relaxivity (slope); the box shows the 95%

confidence bounds of the linear fit. For the R1-iron relaxivity, the inset shows a zoom-out of the main figure, presenting the entire range of measured values. **d** The dependency of R1 on R2* for different iron environments. Data points represent median values of biologically independent samples with varying iron compound concentrations. The linear relationships of R1 and R2* are marked by lines. The slopes of these lines are the $r_1$-$r_2$* relaxivities, which do not require iron concentration estimation and therefore can be estimated in vivo. Dashed lines represent extrapolation of the linear fit. Shaded areas represent the 95% confidence bounds. The inset shows a zoom-out of the main figure, presenting the entire range of measured values (**e**) The $r_1$-$r_2$* relaxivities are different for different iron environments (p(one-sided ANCOVA) = $1.2 \times 10^{-40}$). For each box, the central mark is the $r_1$-$r_2$* relaxivity, and the box shows the 95% confidence bounds of the linear fit. Red dots indicate the successful prediction of the experimental $r_1$-$r_2$* relaxivity from the ratio between the iron relaxivities of R1 and R2* ($\frac{r_{(1,JC)}}{r_{(2,JC)}}$, shown in c, see Eq. 2). This validates our theoretical in vivo relaxivity model.

---

brain[3]. Importantly, these changes were above the detection limit of the in vitro $r_1$-$r_2$* relaxivity measurement (Supplementary Fig. 9). Hemoglobin affects the $R_1$ and $R_2$* relaxivities of blood[39]. In vitro, we found that ferritin and transferrin have distinct $r_1$-$r_2$* relaxivities even in the presence of hemoglobin (Supplementary Section 5). Myelin is a major contributor to $R_1$ and $R_2$* in the brain[4,19,24,28,40–45]. Since myelin is

composed mainly of lipids, we tested the effect of the myelin fraction on the iron relaxivity by varying the liposomal fractions in our in vitro experiments. We found that the $r_1$-$r_2$* relaxivities are stable for different liposomal fractions and lipid types (Supplementary Section 6). These results suggest that the $r_1$-$r_2$* relaxivity is less sensitive to the lipid concentration and composition compared to $R_1$ and $R_2$*.

### The $r_1$-$r_2^*$ relaxivity provides a unique MRI contrast in the in vivo human brain

Following the in vitro validation, we measured the $r_1$-$r_2^*$ relaxivity in the living human brain. For this aim we calculated the linear dependency of $R_1$ on $R_2^*$ across voxels of different anatomically-defined regions of interest (ROIs, see "$r_1$-$r_2^*$ relaxivity computation for ROIs in the human brain" in Methods). We found distinct $r_1$-$r_2^*$ relaxivities for different brain regions (Fig. 3a). The heterogeneous distribution of the in vivo $r_1$-$r_2^*$ relaxivity in the brain was consistent across healthy subjects (age $27 \pm 2$ years, $N = 21$, Fig. 3b) and was reproducible in scan-rescan experiments (Supplementary Fig. 14). Our in vitro results indicated that the $r_1$-$r_2^*$ relaxivity provides different information compared to $R_1$ and $R_2^*$. In agreement, in the in vivo human brain we found that the $r_1$-$r_2^*$ relaxivity produces a unique contrast, statistically different from $R_1$ and $R_2^*$ (Fig. 3b–d; $p = 0.031$ and test statistics $= 0.13$ for the two-sample Kolmogorov–Smirnov test comparing the $r_1$-$r_2^*$ relaxivity distribution to $R_1$, and $p = 2.5 \times 10^{-4}$ and test statistics $= 0.19$ comparing it to $R_2^*$). While the $r_1$-$r_2^*$ relaxivity represents the slope of the $R_1$-$R_2^*$ linear fit, we also find that the intercept varies across the brain (Supplementary Section 7).

The $r_1$-$r_2^*$ relaxivity is calculated for an anatomically-defined ROI in the brain, in which the fit is performed across voxels. A voxel-wise $r_1$-$r_2^*$ relaxivity visualization based on each voxel's local neighborhood, as well as comparison to the $R_1$ and $R_2^*$ contrasts, is demonstrated in Supplementary Section 8.

### The effect of myelin on the $r_1$-$r_2^*$ relaxivity

The sensitivity of $R_1$ and $R_2^*$ to the myelin content is known to produce contrasts that are governed mainly by the differences between white-matter and gray-matter tissues[4,19,24,28,40–44].

As expected, we find a strong distinction between gray-matter and white-matter regions in $R_1$ and $R_2^*$ values (Fig. 3c, d). However, the contrast of the $r_1$-$r_2^*$ relaxivity across the brain shows a unique spatial pattern and reveals differences between brain regions beyond the typical white matter−gray matter differentiation (Fig. 3b). For example, we found the temporal, parietal and occipital white-matter regions to be indistinguishable in terms of their $R_1$ and $R_2^*$ values (p(one-way ANOVA) $= 0.99$, statistics(2) $= 0.01$ for R2* and p(one-way ANOVA) $= 0.48$, statistics(2) $= 0.74$ for R1, Supplementary Fig. 20), but these regions were separable based on their different $r_1$-$r_2^*$ relaxivities (p(one-way ANOVA) $= 5 \times 10^{-11}$, statistics(2) $= 36$, Supplementary Fig. 20).

To further establish that the $r_1$-$r_2^*$ relaxivity is less sensitive to the myelin content relative to $R_1$ and $R_2^*$ we compared it to several in vivo myelin markers. The qMRI measurements of the macro-molecular tissue volume (MTV)[46], magnetization transfer saturation (MTsat)[47], and mean diffusivity (MD) were shown to approximate the myelin content and characteristics[48–53]. These myelin-sensitive markers were highly correlated with $R_1$ and $R_2^*$ but were not significantly correlated with the $r_1$-$r_2^*$ relaxivity (Fig. 4a, Supplementary Figs. 21, 22). In a biophysical model of the $r_1$-$r_2^*$ relaxivity, that accounts for the presence of myelin and iron compounds, we find that the variability of myelin within an ROI can affect the $r_1$-$r_2^*$ relaxivity measurement (Supplementary Section 9). Yet, an in vivo estimate of this myelin characteristic revealed it explains only 30% of the variability in the in vivo $r_1$-$r_2^*$ relaxivity across the brain (Supplementary Fig. 28). We also performed a set of numerical simulations in which we consider the contributions of multiple brain tissue components to the relaxivity measurement (Supplementary Section 9). As in the in vivo brain, we found that changes in the myelin concentration substantially affect the simulated measurements of $R_1$, $R_2^*$. However, myelin-related changes were not the main component governing the simulated measurement of the $r_1$-$r_2^*$ relaxivity, and in simulations of physiological conditions they could not explain the variability in the $r_1$-$r_2^*$ relaxivity across the brain. We

also find that T1w/T2w contrast, which serves as a semi-quantitative myelin marker[54], is different from the $r_1$-$r_2^*$ relaxivity (Supplementary Section 10). Importantly, the $R_1$/$R_2^*$ ratio may be more similar to the $r_1$-$r_2^*$ relaxivity than the individual measurements of R1 and R2*. Yet, in vivo estimation and numerical simulation of the $R_1$/$R_2^*$ ratio show it is uncorrelated with the $r_1$-$r_2^*$ relaxivity and has a different biophysical interpretation (Supplementary Section 11).

### The $r_1$-$r_2^*$ relaxivity correlates with the iron mobilization capacity across the brain and in aging

Next, we tested the sensitivity of the $r_1$-$r_2^*$ relaxivity to the state of iron homeostasis across the normal brain and in aging. We aggregated previously reported postmortem histological data describing iron, ferritin and transferrin concentrations in different brain regions of young (aged 27–64 years, $N > = 7$) and older (aged 65–88 years, $N > = 8$) adults[5,7,9] (see "Group-level comparison of qMRI parameters and histological measurements" in Methods). We performed a group-level comparison between these postmortem findings and in vivo MRI parameters, which we measured in the same brain regions and age groups (healthy young subjects aged 23–63 years, $N = 26$; older subjects aged 65–77 years, $N = 13$). As expected, $R_2^*$ was significantly correlated with iron concentration ($R^2 = 0.56$, p(one-sided F-test corrected for FDR) $= 2.8 \times 10^{-4}$, statistics(20) $= 25.4$; Fig. 4b, full statistics are in Supplementary table 2). Importantly, this result validates the agreement between the in vivo and postmortem datasets, thus allowing to further examine the biological correlates of the $r_1$-$r_2^*$ relaxivity. For this aim we estimated a feature of the iron homeostasis, the iron mobilization capacity (transferrin/iron[7]), available in the postmortem dataset. This measure was not correlated with $R_2^*$ or $R_1$ (Fig. 4c). However, the iron mobilization capacity[5,7] was significantly correlated with the $r_1$-$r_2^*$ relaxivity across brain regions and age groups ($R^2 = 0.62$, p(one-sided F-test corrected for FDR) $= 7.3 \times 10^{-5}$, statistics(20) $= 32.7$, Fig. 4c). Hence, the $r_1$-$r_2^*$ relaxivity, unlike $R_1$ and $R_2^*$, is sensitive to the obtained ex vivo measurements of the iron mobilization capacity across the brain and can capture the effect of aging on this iron homeostasis marker.

The $r_1$-$r_2^*$ relaxivity was much less correlated with the absolute ferritin, transferrin or iron concentrations (Fig. 4b, Supplementary Fig. 38). Moreover, the correlation with the iron concentration was driven mostly by outliers (the globus pallidum), unlike the sensitivity of the $r_1$-$r_2^*$ relaxivity to the iron homeostasis marker (Supplementary Section 12). Therefore, the $r_1$-$r_2^*$ relaxivity is less sensitive to absolute changes in the iron concentrations. In return, these results imply that the $r_1$-$r_2^*$ relaxivity enhances the sensitivity of MRI to the iron homeostasis. We corroborated these findings with numerical simulations of the $r_1$-$r_2^*$ relaxivity (Supplementary Section 9).

### The $r_1$-$r_2^*$ relaxivity enhances the distinction between tumor tissues and non-pathological tissue

While the $r_1$-$r_2^*$ relaxivity forms a unique pattern of changes across the brain, it needs to be established that this contrast contains meaningful clinical information, that can complement the contrasts of $R_1$ and $R_2^*$. For this aim, we evaluated the MRI contrast between pathological and normal-appearing tissues of patients with meningioma brain tumors ($N = 18$, Fig. 5a, b, full statistics are in Supplementary Table 3). The diagnosis of brain tumors and their delineation from the surrounding non-pathological tissue is routinely performed using contrast-enhanced MRI, which requires the injection of an external gadolinium (Gd)-based contrast agent with paramagnetic properties[55]. As expected, when using Gd-based contrast, tumor tissue was distinct from white-matter and gray-matter tissues (Fig. 5c, Cohen's $d = 1.4$, $p < 10^{-4}$, statistics(16) $= -5.8$ for tumor-gray matter; Cohen's $d = 1.18$, $p < 10^{-3}$ statistics(16) $= -4.9$ for tumor-white matter). Recently renewed concerns about the long-term safety of Gd-based agents[56,57], highlight the need for Gd-free MRI techniques that can serve as safe

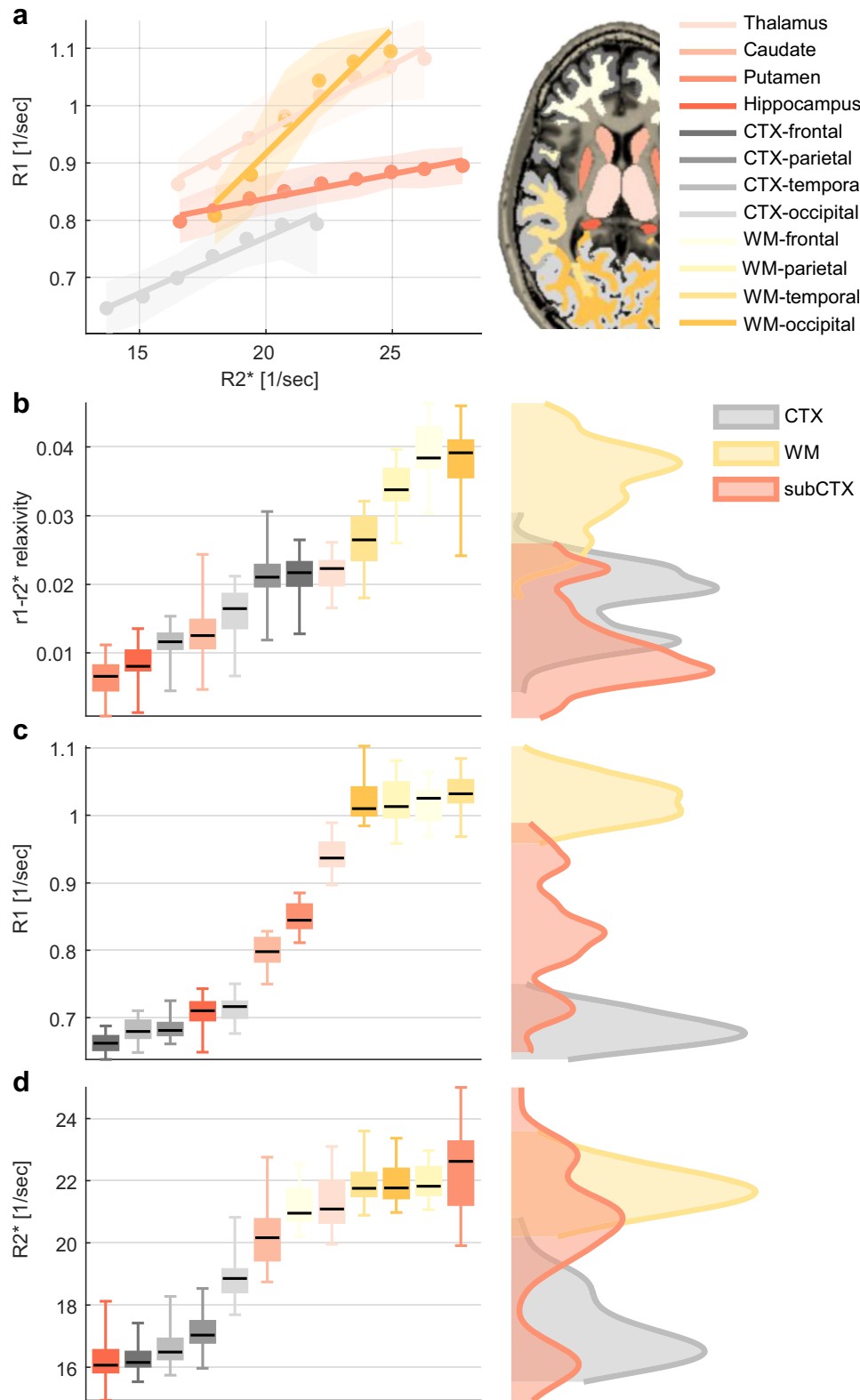

**Fig. 3 | The in vivo $r_1$-$r_2$\* relaxivity provides a unique contrast in the brain. a** The dependency of R1 on R2* in four representative brain regions; occipital white matter (WM-occipital), occipital cortex (CTX-occipital), Thalamus & Putamen of a single subject. R2* and R1 measurements across voxels were binned (dots represent the median; shaded areas represent the mean absolute deviation), and a linear fit was calculated. The slopes of the linear fit represent the dependency of R1 on R2* ($r_1$-$r_2$\* relaxivity) and vary across brain regions. **b** The $r_1$-$r_2$\* relaxivity across the brain. Left: the $r_1$-$r_2$\* relaxivity in different brain regions, the variation in each region is across normal subjects (age $27 \pm 2$, $N = 21$). Within-region, this measurement is

stable across subjects. It shows clear difference between regions, thus indicating its reliability. The 25th, 50th, and 75th percentiles and extreme data points are shown for each box. Right: the contrast of the $r_1$-$r_2$\* relaxivity across the brain. Red, yellow and gray distributions represent the values of the $r_1$-$r_2$\* relaxivities in sub-cortical (sub-CTX), white-matter (WM) and cortical (CTX) brain regions, respectively.
**c**, **d** Similar analyses for R1 and R2* values, in which the gray-matter vs. white-matter contrast is much more dominant compared to the $r_1$-$r_2$\* relaxivity. Hence, the $r_1$-$r_2$\* relaxivity provides distinct information compared to R1 and R2*, beyond the WM-GM separation. Results in this entire figure are for ROIs in the left hemisphere.

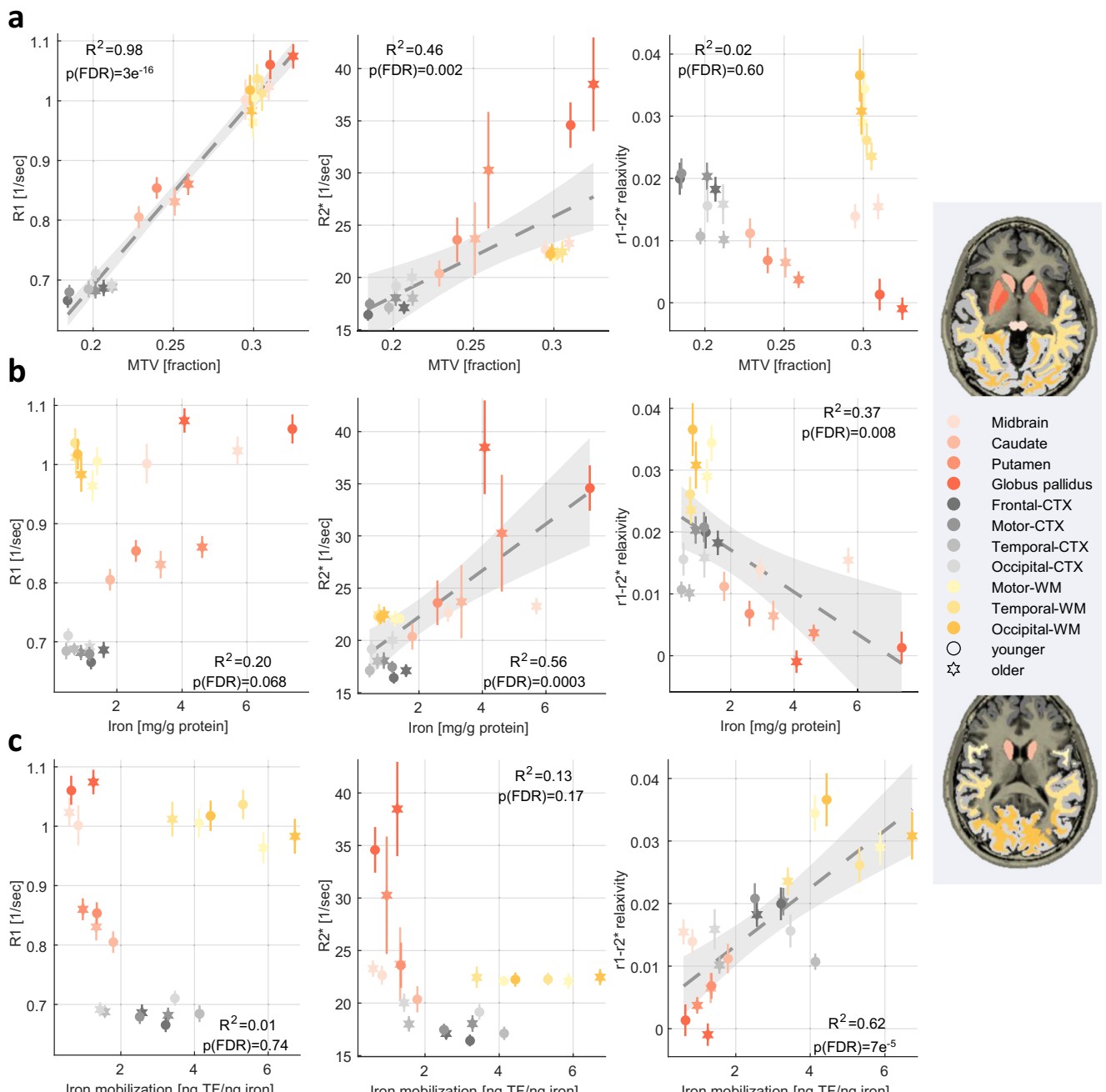

**Fig. 4 | The $r_1$-$r_2^*$ relaxivity has unique biological correlates compared to R1 and R2* across the brain and in aging.** R1, R2* and the $r_1$-$r_2^*$ relaxivity were measured in vivo across younger (aged 23–63 years, $N = 26$) and older (aged 65–77 years, $N = 13$) subjects (different marker shapes) in 11 brain regions (different colors, WM white matter, CTX cortex). Each row presents the correlations of these qMRI measurements with a different in vivo or ex vivo histological feature (fitted model and 95% confidence bounds are presented for significant correlations): (**a**) qMRI vs. the macromolecular tissue volume (MTV), an in vivo myelin-sensitive marker, measured for younger (aged 23–63 years, $N = 26$) and older (aged 65–77 years, $N = 13$) subjects. Unlike R1 and R2*, the $r_1$-$r_2^*$ relaxivity is not linearly related to this myelin-sensitive marker (See Supplementary Figs. 21, 22 for additional in vivo myelin-sensitive markers). **b** qMRI vs. the iron concentration (postmortem, from

the literature[5,7]) measured for younger (aged 27–64 years, $N >= 7$) and older (aged 65–88 years, $N >= 8$) subjects (for precise sample sizes see Table 2). Notably, the $r_1$-$r_2^*$ relaxivity is not significantly correlated with the iron content when excluding the outlier values in the globus pallidus while the R2* correlation survives this exclusion (Supplementary Section 12). **c** qMRI vs. the iron mobilization capacity (transferrin (TF) /iron ratio), an iron homeostasis marker (postmortem, from literature[5,7], same subjects as in **b**). Only the $r_1$-$r_2^*$ relaxivity is significantly correlated with the iron mobilization capacity, implying for its sensitivity to the iron homeostasis across brain regions and in aging. For all panels, data points show mean values and error bars show the mean absolute deviation across subjects, $p$-values are for one-sided F-test corrected for multiple comparisons (FDR), full statistics are in supplementary table 2.

alternatives[58]. However, without Gd-agent injection, both for $R_1$ and $R_2^*$ values the biggest effect size was observed between white-matter and gray-matter tissues, with no significant difference between gray-matter and tumor tissues (Cohen's $d = 0.43$, $p = 0.08$, statistics(17) = −1.8 for $R_1$, Cohen's $d = 0.10$, $p = 0.67$, statistics(17) = 0.43 for $R_2^*$, Fig. 5d, e).

This demonstrates the poor performances of $R_1$ and $R_2^*$ in Gd-free tumor tissue delineation. Importantly, the $r_1$-$r_2^*$ relaxivity greatly enhanced the contrast between tumor tissue and non-pathological tissue, without contrast agent injection (Fig. 5f, Cohen's $d = 1.5$, p = 7 × $10^{-6}$, statistics(17) = −6.4 for tumor-gray matter; Cohen's $d = 4.32$,

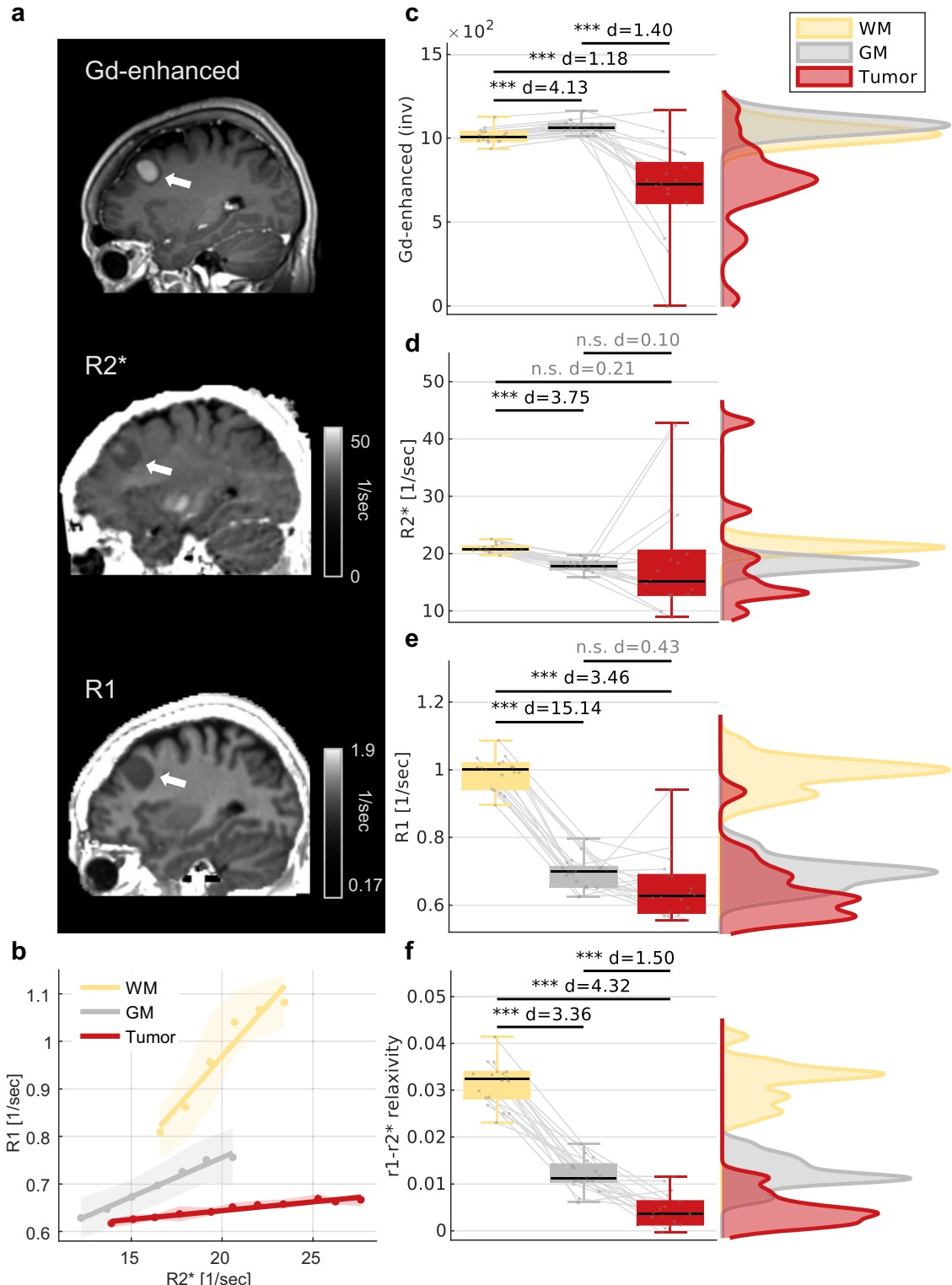

$p = 1.2 \times 10^{-12}$, statistics(17) = −18.3 for tumor-white matter).

This Gd-free enhancement was comparable in size to the effect of Gd-based contrast. Moreover, we provide an example for a voxel-wise visualization of the $r_1$-$r_2^*$ relaxivity approach in a representative meningioma patient, which demonstrates the Gd-free tumor enhancement (Supplementary Fig. 18). These results emphasize the improved sensitivity of the $r_1$-$r_2^*$ relaxivity to the unique tumor microenvironment, which may have wide clinical implications as a safe alternative for contrast agents' injections.

## The $r_1$-$r_2^*$ relaxivity is associated with unique biological pathways and gene expression profiles

We further examined how the tumor characteristics obtained by the $r_1$-$r_2^*$ relaxivity differ from the information contained in $R_1$ and $R_2^*$. For this aim, we examined the associations of in vivo MRI measurements with underlying gene-expression profiles for the same tissue. We analyzed cases in which the MRI scans of the meningioma patients were followed by surgical interventions, to obtain matching resected tumor tissue samples that we profiled by bulk RNA-sequencing

**Fig. 5 | Application of the $r_1$-$r_2^*$ relaxivity on meningioma brain tumors. a** From top to bottom: Gd-enhanced T1-weighted image, R1 map and R2* map in a representative subject with a meningioma brain tumor (white arrow). **b** The dependency of R1 on R2* (the $r_1$-$r_2^*$ relaxivity) for the white-matter (WM, frontal), gray-matter (GM, frontal) and tumor tissue of the same subject. R2* and R1 measurements across voxels were binned (dots represent the median; shaded areas represent the mean absolute deviation), and a linear fit was calculated. The slopes of the linear fit represent the $r_1$-$r_2^*$ relaxivity. Tumor tissue exhibits distinct $r_1$-$r_2^*$ relaxivity relative to non-pathological tissue. The $r_1$-$r_2^*$ relaxivity is calculated across voxels (R1 and R2* values shown in a) for each ROI. A voxel-wise visualization of the $r_1$-$r_2^*$ relaxivity available in Supplementary section 8. **c**–**f** The contrast between the white-matter (WM), gray-matter (GM) and tumor tissues across patients for the Gd-enhanced

contrast (inverted and scaled relative to the maximal value across subjects for visualization, [a.u.], $N = 17$) (**c**), R2* ($N = 18$) (**d**), R1 ($N = 18$) (**e**) and the $r_1$-$r_2^*$ relaxivity ($N = 18$) (**f**). Only the $r_1$-$r_2^*$ relaxivity produces significant differences between tumor and GM tissues without contrast agents. Left: boxes present the variation in the contrasts across patients. The 25th, 50th, and 75th percentiles and extreme data points are shown. The $d$-values represent the effect size (Cohen's $d$) of the differences between tissue types, and the significance level is based on a paired-sample $t$ test (two-sided). Gray lines extend between values of the same patient. Right: the distribution of the values between WM, GM and tumor tissue across patients. Estimates in non-pathological tissues are for the tumor-free hemisphere. $p < 0.05$; **$p < 0.01$; ***$p < 0.001$ (full statistics are in supplementary table 3).

(Fig. 6a). For these tumor samples ($N = 17$), we performed an unbiased analysis to identify genes and molecular pathways that could be linked with the in vivo measured MRI parameters. For each gene we calculated the correlation between the expression level and the in vivo MRI measurements ($r_1$-$r_2^*$ relaxivity, $R_1$ and $R_2^*$) across patients. We then performed gene set enrichment analysis (GSEA)[59,60] to identify molecular functions that are significantly associated with each MRI measurement. In total, we found 9, 55 and 59 significantly enriched gene sets for $R_1$, $R_2^*$ and the $r_1$-$r_2^*$ relaxivity, respectively (p(two-sided permutation test) < 0.01 after familywise error rate (FWER) correction; Supplementary Table 4 and Supplementary Fig. 40). These gene sets define genes linked to a specific biological pathway. Almost half of the significant gene sets were exclusively associated with the $r_1$-$r_2^*$ relaxivity, and not with $R_1$ or $R_2^*$ (Supplementary Fig. 40). The enrichment score represents the degree to which the genes within a set were positively or negatively correlated with MRI measurements. In examining the associations of MRI measures to biological pathways, as reflected in the enrichment score, we found that the $r_1$-$r_2^*$ relaxivity clustered separately from $R_1$ and $R_2^*$ (Fig. 6b). The clustering results were replicated when performed on the p-value of the enrichment, or on the subset of genes within the top enrichment pathways. This implies that the $r_1$-$r_2^*$ relaxivity reflects unique cellular and molecular properties, undetectable by the separate analysis of $R_1$ and $R_2^*$. Therefore, the in vivo $r_1$-$r_2^*$ relaxivity provides a unique dimension for measuring microstructure and gene expression features across the brain. The gene enrichment analysis that we performed on resected brain tumors (Fig. 6b) can provide insights into the biological pathways associated with the $r_1$-$r_2^*$ relaxivity. The two most enriched pathways for the $r_1$-$r_2^*$ relaxivity were "immunoglobulin complex" (normalized enrichment score (NES) = −3.62, FWER $p$-value < 0.001; Supplementary Fig. 41a) and "scavenging of heme from plasma" (NES = −3.27, FWER $p$-value < 0.001; Supplementary Fig. 41b). While the former may relate to the response of the immune system to the cancerous process[61,62], the latter involves the absorption of free heme, a source of redox-active iron[63]. This iron-related pathway was not significantly associated with $R_1$ (FWER $p$-value > 0.05) and was less significantly associated with $R_2^*$ compared to the $r_1$-$r_2^*$ relaxivity (FWER $p$-value = 0.038). Moreover, we examined the main genes involved in iron regulation: transferrin receptor (TFRC), ferritin heavy-chain polypeptide 1 (FTH1) and ferritin light-chain polypeptide (FTL)[64,65]. Both TFRC and FTH1 were included in the subset of genes within the top enrichment pathways for the $r_1$-$r_2^*$ relaxivity, but were not found to be associated with $R_1$ or $R_2^*$. These findings provide evidence at the level of gene expression for the sensitivity of the $r_1$-$r_2^*$ relaxivity to iron homeostasis.

### The $r_1$-$r_2^*$ relaxivity reveals differences in iron homeostasis between tumor tissues

We further validated the sensitivity of the $r_1$-$r_2^*$ relaxivity to the iron homeostasis at the proteomics level, by comparing in vivo MRI measurements to ex vivo iron homeostasis estimation on the same tissue (Fig. 6a). MRI scans of the meningioma patients were followed by

surgical interventions, to obtain matching resected tumor tissue samples that we analyzed by western-blot. We compared in vivo MRI values of tumor tissue to its transferrin/ferritin ratio which serves as a proxy for iron homeostasis. Neither $R_1$ nor $R_2^*$ showed significant differences between tumors with low and high transferrin/ferritin ratios (Fig. 6c, $p = 0.84$, statistics(14) = −0.2 for R1 and $p = 0.93$, statistics(14) = −0.1 for R2*). Notably, the $r_1$-$r_2^*$ relaxivity was significantly higher for tumors with high transferrin/ferritin ratio compared to tumors with low transferrin/ferritin ratio ($p = 0.005$, statistics(14) = −3.3, Fig. 6c). Therefore, as established by both gene expression and proteomics analyses, the $r_1$-$r_2^*$ relaxivity measured in vivo detects pathological disruptions in iron homeostasis which were previously only observable ex vivo.

## Discussion

We present an MRI relaxivity approach, with increased sensitivity to different molecular environments of iron. First, we confirm in vitro that different iron environments induce different relaxivities, which can be estimated with MRI using the $r_1$-$r_2^*$ relaxivity. When examining $R_1$ and $R_2^*$ of different iron environments, we find the molecular state of iron is confounded by the strong effects of iron and myelin concentrations. In this in vitro setting, we show that the $r_1$-$r_2^*$ relaxivity reveals the sensitivity of MRI to the intrinsic paramagnetic properties of different iron environments. In the healthy human brain, we show that the $r_1$-$r_2^*$ relaxivity provides a unique MRI contrast. This contrast is less sensitive to myelin compared to other quantitative MRI parameters. Interestingly, it does vary with an iron homeostasis marker[7] across brain regions and age groups. We further demonstrate that the $r_1$-$r_2^*$ relaxivity contains meaningful clinical information associated with iron homeostasis, which was previously inaccessible to conventional MRI approaches. In meningioma patients, we find that the $r_1$-$r_2^*$ relaxivity contrast is useful for enhancing the distinction between tumor tissue and non-pathological tissue. We substantiate this finding by associating in vivo MRI measurements with RNA sequencing and protein expression levels of the same tumor tissues. We find that the $r_1$-$r_2^*$ relaxivity is associated with unique iron-related biological pathways and reveals the state of iron homeostasis in tumors.

Relaxivity is commonly employed to characterize MR contrast agents[66]. While most contrast agents induce relaxation based on their paramagnetic or superparamagnetic properties, some agents elevate the $R_1$ relaxation rate more efficiently while others elevate $R_2^*$. $R_1$ relaxation mechanisms are affected by local molecular interactions, while $R_2^*$ is sensitive to more global effects of extended paramagnetic interactions at the mesoscopic scale[36]. We show that by contrasting these two different mechanisms, we gain sensitivity to the endogenous iron environment, without the injection of an external contrast agent.

The concept of iron relaxivity, and its sensitivity to the molecular environment of iron, was previously suggested by several postmortem and in vitro studies[30–34]. We reproduce these results in our in vitro experiments and further demonstrate that different iron environments have different iron relaxivities. Moreover, Ogg et al.[30] calculated the iron relaxivity by comparing postmortem measurements of iron

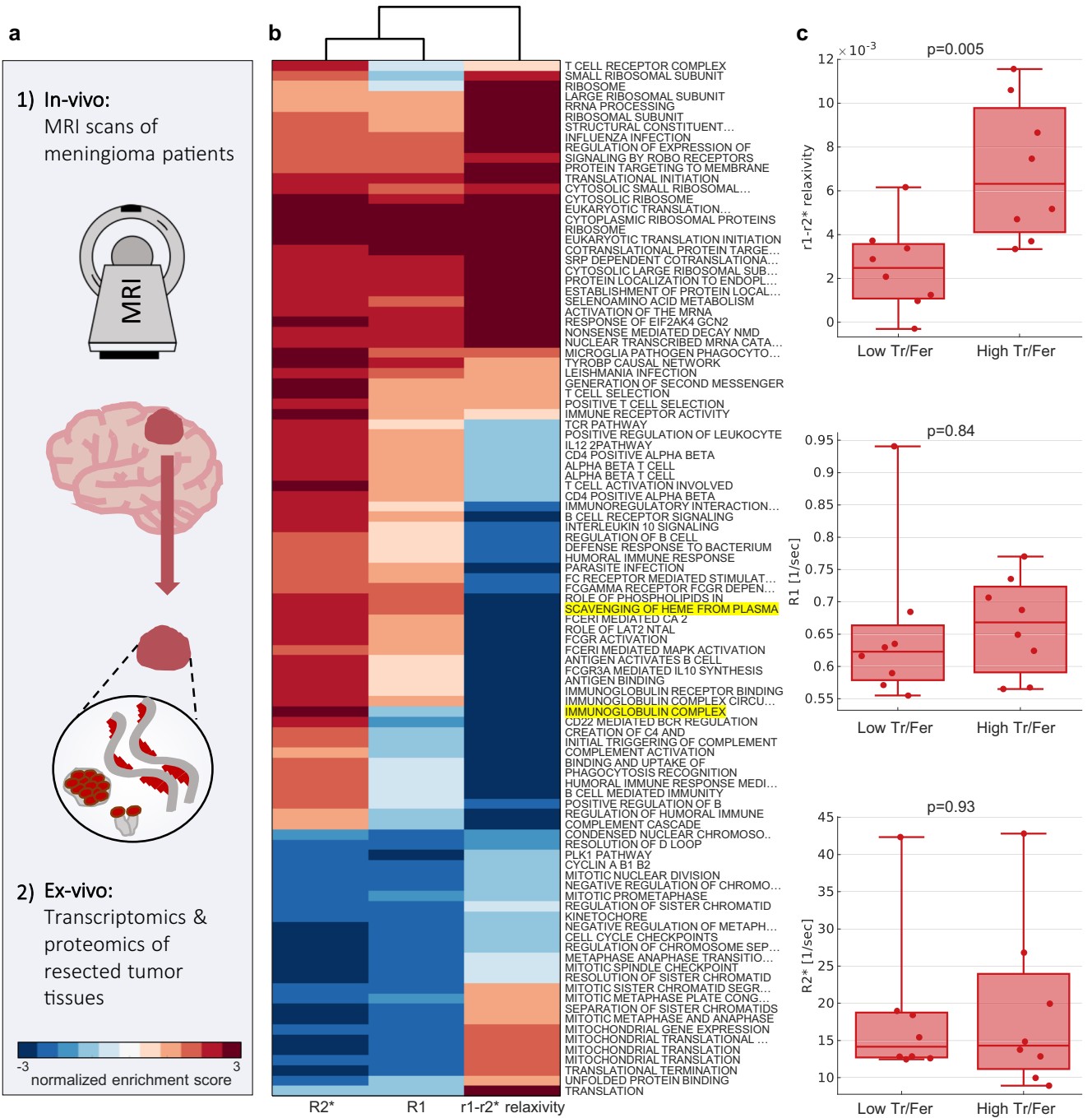

**Fig. 6 | The $r_1$-$r_2$* relaxivity measured in vivo in meningioma patients agrees with iron homeostasis markers estimated ex vivo on surgical specimens of meningiomas from the same patients. a** Unique comparison between qMRI parameters measured in vivo and ex vivo iron histology of the same human tissue; MRI scans of meningioma patients ($N = 17$) were followed by surgical interventions, to obtain matching resected tumor tissue samples. These tissue samples were profiled by bulk RNA-sequencing to obtain gene expression profiles, and by western-blot analysis to estimate the transferrin/ferritin ratio, a marker for iron homeostasis. **b** Gene set enrichment analysis for the correlation of MRI with gene expression. Rows show significant biological pathways, columns represent R1, R2* and the the $r_1$-$r_2$* relaxivity. The dendrogram shows hierarchical clustering of the normalized enrichment scores. The $r_1$-$r_2$* relaxivity clustered separately from R1 and

R2* and is therefore enriched for unique biological pathways. The two most enriched pathways for the $r_1$-$r_2$* relaxivity are highlighted in yellow. **c** The $r_1$-$r_2$* relaxivity, R1 and R2* measured in vivo for tumor tissues classified as having either low or high transferrin-to-ferritin ratios (Tf/Fer). Tf/Fer ratio, an iron homeostasis marker, was estimated using western-blot analysis following surgical resection of the tissue. Data points represent independent measurements of individual patients ($N = 16$). The threshold between groups was set to 1 based on the median across patients ($N = 8$ for each group). While R1 and R2* cannot distinguish between the groups with different Tf/Fer ratios, the $r_1$-$r_2$* relaxivity is higher in tissues with a high Tf/Fer ratio, indicating its sensitivity to iron homeostasis. Each box shows the 25th, 50th, and 75th percentiles and extreme data points. p-values presented are for two-sample *t* tests (two-sided).

concentration for different age groups to the typical $R_1$ values in those age groups. They found that this approximation of iron relaxivity was higher in the gray matter and white matter than in sub-cortical structures. We replicate this result in living subjects, based on our approach

for estimating iron relaxivity in vivo. The theoretical derivation we propose for the $r_1$-$r_2$* relaxivity shows that it represents the ratio of the iron relaxivities of $R_1$ and $R_2$*. This theory was supported by our in vitro experiments. Therefore, we exploit the different relaxation rates for a

biophysical model of their linear interdependency, thus allowing to approximate the iron relaxivity in the living brain.

First, we evaluated the sensitivity of the $r_1$-$r_2^*$ relaxivity to the molecular environment of iron based on in vitro experiments with ferritin, transferrin, and ferrous iron ions. We show that these iron environments induce different relaxivities, even when accounting for the discrepancies in iron binding. Moreover, we show that the macromolecular environment in which the iron reside can alter the relaxivity as well. For example, ferritin induces different relaxivities in the presence of liposomes, proteins or when it is free. As implied previously[31], this effect can be explained by the encapsulation and the spatial distribution of paramagnetic molecules. Therefore, similarly to different contrast agents, the physical mechanisms by which iron compounds interact with the surrounding water environment are inherently different. To further confirm that our results are related to paramagnetic properties and not the presence of the proteins themself, we tested MRI measurements of apo-transferrin (transferrin unbound to iron). In this case, the $r_1$-$r_2^*$ relaxivity of transferrin vanished. Therefore, the presence of iron in different in vitro environments can be detected by the $r_1$-$r_2^*$ relaxivity. Moreover, evaluating ferritin-transferrin mixtures, we find that in heterogenous iron environments the specific composition of the iron milieu affects the $r_1$-$r_2^*$ relaxivity.

The biological interpretation of the $r_1$-$r_2^*$ relaxivity measurement is more ambiguous when applied to the in vivo human brain. This is due to the fact that most qMRI parameters, including $R_1$ and $R_2^*$, are known to suffer from low biological specificity in the brain[4,19,24,28,40–42]. The primary MRI contrast between gray matter and white matter usually is associated with myelin, while an additional and often correlated effect is attributed to the iron concentration[4]. A major strength of our relaxivity approach is that it captures distinct biological and pathological information, not acquired by traditional qMRI parameters such as $R_1$ and $R_2^*$. We show that the great contrast between white matter and gray matter usually observed in $R_1$ and $R_2^*$ is no longer as substantial in the $r_1$-$r_2^*$ relaxivity. In vitro, we show that the $r_1$-$r_2^*$ relaxivity is stable across iron and liposomal concentrations. To further demonstrate the minimal effect of myelin on the $r_1$-$r_2^*$ relaxivity in the brain, we employed the qMRI measurements of MTV[46], MTsat[47] and MD, which were shown to approximate the myelin content[48–53]. These in vivo myelin markers are all highly correlated with $R_1$ and $R_2^*$ but not with the $r_1$-$r_2^*$ relaxivity. The semi-quantitative T1w/T2w myelin marker[54] is also not correlated with the $r_1$-$r_2^*$ relaxivity. Adapting the biophysical model for the $r_1$-$r_2^*$ relaxivity to account for myelin, we find that the variability in myelin within an ROI can still affect the $r_1$-$r_2^*$ relaxivity measurement. Yet, in vivo estimate of the myelin variability only explains ~30% of the variation in the in vivo $r_1$-$r_2^*$ relaxivity across the brain. In brain tissue numerical simulations, we show that the myelin content substantially affects the measurements of $R_1$ and $R_2^*$, but it cannot by itself explain the measured variability in the $r_1$-$r_2^*$ relaxivity across the brain. The $R_1$/$R_2^*$ ratio may be more similar to the $r_1$-$r_2^*$ relaxivity than the individual measurements of $R_1$ and $R_2^*$. However, the $R_1$/$R_2^*$ ratio is sensitive to the relative magnitudes of $R_1$ and $R_2^*$, and the $r_1$-$r_2^*$ relaxivity is only affected by the shared variation of $R_1$ and $R_2^*$. We demonstrate in vivo that these measurements are different. In simulations we find that unlike the $r_1$-$r_2^*$ relaxivity, the $R_1$/$R_2^*$ ratio is sensitive to iron compounds concentration and myelin content. Nonetheless, it could be that the $r_1$-$r_2^*$ relaxivity contains some residual contribution of the myelin content. We show that this contribution is much more limited compared to traditional qMRI parameters such as $R_1$ and $R_2^*$. In return, the MRI measurement of the $r_1$-$r_2^*$ relaxivity enhances the contrast between pathological tissue and normal tissue and is associated with distinct gene expression pathways. Hence, the relaxivity framework reveals distinct biological features otherwise undetectable in standard qMRI measurements.

Still, an open question remains whether the biological interpretation of the $r_1$-$r_2^*$ relaxivity demonstrated in vitro is also measurable in vivo. We bring several evidence that the $r_1$-$r_2^*$ relaxivity reveals the sensitivity of MRI to properties of the molecular iron environment, otherwise confounded by myelin and iron concentrations. First, in the healthy and aging brain we show that unlike $R_1$ and $R_2^*$, the $r_1$-$r_2^*$ relaxivity is correlated with an iron homeostasis marker, the iron mobilization capacity (transferrin/iron). We also show that the $r_1$-$r_2^*$ relaxivity is less correlated with the absolute ferritin, transferrin or iron concentrations, indicating that it enhances the sensitivity to the homeostasis between iron compounds, rather than their absolute concentrations. Importantly, we corroborate these findings based on in vivo and ex vivo analyses of meningioma patients. We find that the variability in the in vivo $r_1$-$r_2^*$ relaxivity of brain tumors is explained by their transferrin-ferritin ratios and is associated with iron-related genes. These results could not be obtained for the individual measurements of $R_1$ and $R_2^*$. To further validate that the $r_1$-$r_2^*$ relaxivity could be sensitive to the iron homeostasis, even when accounting for the massive effect of the myelin and iron concentrations on the MRI signal, we generated a simulation of a brain-like environment which contains multiple tissue. We provide an example for physiological changes in the iron environment, based on the ferritin-transferrin fraction, which could lead to considerable changes in the $r_1$-$r_2^*$ relaxivity, well above the detection limit of this MRI measurement. Therefore, we show in vitro, in vivo, ex vivo and in numerical simulations, that the $r_1$-$r_2^*$ relaxivity measurement could allow to detect changes in iron homeostasis under physiological conditions.

Our in vitro experiments are based on ferritin, transferrin and ferrous iron ions as examples for variable iron environments. Yet, in the human brain our approach is probably more broadly sensitive to the entire milieu of iron compounds, their iron binding characteristics and aggregation states. Other iron compounds that exist in the brain, such as hemoglobin, hemosiderin, neuromelanin, magnetite, ferric ion, lactoferrin and melanotransferrin[4], might have distinct iron relaxivities as well (as we have shown for the iron relaxivity of hemoglobin). Moreover, other characteristics of the iron environment such as iron compounds' aggregate sizes, intra-aggregate spacing, spatial distributions and iron loadings, could all have an additional effect on the iron relaxivity[34]. For example, catecholamine neurons of the substantia nigra and locus coeruleus are rich in neuromelanin-iron complexes[4,67] which could contribute to the $r_1$-$r_2^*$ relaxivity measurement in these regions (as evident by our distinct results in the pallidum). In addition, the paramagnetic properties of deoxyhemoglobin in capillaries and veins and their orientations are known to affect the $R_2^*$ measurement[68–70]. In vitro, we found that ferritin and transferrin have distinct $r_1$-$r_2^*$ relaxivities even in the presence of hemoglobin. However, hemoglobin could have a more substantial effect in vivo. Thus, the $r_1$-$r_2^*$ relaxivity might also be sensitive to the iron environment in the vascular system, which is crucial for brain iron metabolism and homeostasis[71]. Moreover, on the cellular scale there is substantial heterogeneity in the iron environment[72,73]. $R_1$ and $R_2^*$ each represent some average of different spatial and cellular compartments, but it is unclear how the $r_1$-$r_2^*$ relaxivity is weighting such cellular contributions. Similar to the measurements of $R_1$ and $R_2^*$, the $r_1$-$r_2^*$ relaxivity might be contaminated by biases related to magnetic field inhomogeneities[23]. Yet, we show that the $r_1$-$r_2^*$ relaxivity is reproducible in scan-rescan experiments (both in vitro and in vivo) and is stable across subjects, thus indicating the limited effect of such biases. Another limitation of our approach is that we estimate the dependency of R1 on R2* across voxels, and it therefore produces one relaxivity measurement per anatomically-defined ROI. To visualize the contrast of the $r_1$-$r_2^*$ relaxivity in the brain, we present a voxel-wise implementation based on a sliding-window approach. However, this implementation is more sensitive to partial volume and smoothing effects. To conclude, while we demonstrate our approach in vitro based on ferritin, transferrin and ferrous iron samples, we believe that in the human brain the $r_1$-$r_2^*$ relaxivity provides more comprehensive information on the molecular iron milieu previously inaccessible by MRI.

In order to further model the separate contributions to the MRI signal of different iron compounds, it would be necessary to increase the dimensionality of the in vivo iron relaxivity measurement. The intercept of the $R_1$-$R_2^*$ linear fit represents the residual $R_1$ not explained by $R_2^*$. We find that the relaxivity and the intercept contribute complementary information both in vivo and in vitro. Therefore, combining these measurements may allow better in vivo characterization of brain tissue. In addition to $R_1$ and $R_2^*$, other qMRI parameters known to be sensitive to iron include quantitative susceptibility mapping (QSM) and R2[3,4]. In addition, it was suggested that magnetization transfer (MT) measurements are affected by neuromelanin–iron complexes[12,74]. The linear interdependencies of these other iron-related MRI measurements may uncover additional features of the iron environment[75]. For example, we show that the dependency of $R_1$ on $R_2$ is also useful for differentiating between iron environments. Therefore, we speculate that the concept we introduce here, of exposing the iron relaxivities in vivo based on the linear dependency of $R_1$ on $R_2^*$ (the $r_1$-$r_2^*$ relaxivity), can be generalized to further increase MRI's specificity for iron using additional complementary measurements. In a previous work we implemented a different aspect of relaxivity for the detection of lipid composition, based on the linear dependency of qMRI parameters on the macromolecular tissue volume (MTV)[46]. Here, we demonstrate that the $r_1$-$r_2^*$ relaxivity and the dependency of $R_1$ on MTV provide two orthogonal microstructural axes. The dependency of $R_1$ on MTV changes according to the lipid types, even in the presence of iron, while the $r_1$-$r_2^*$ relaxivity is insensitive to the lipid types and provides better distinctions between iron environments. Therefore, the presented framework can be generalized further to boost MRI's specificity and support a more comprehensive in vivo histology with qMRI.

While the field of in vivo histology with MRI is rapidly growing, ground-truth validation remains a great challenge. Here we propose a cutting-edge validation strategy combining both bottom-up and top-down approaches in which we incorporate in vitro, in vivo and ex vivo analyses. For the bottom-up analysis, we developed a unique, synthetic biological system that allows us to examine the biophysical interpretation of the $r_1$-$r_2^*$ relaxivity in highly controlled in vitro settings. For the top-down analysis, we tested whether our interpretation remains valid in the context of the highly complex biological tissue. We compared the $r_1$-$r_2^*$ relaxivity measured in vivo to histological measurements of iron homeostasis and gene expression. This comparison was done both at the group level, based on previously reported findings, and at the single-subject level, by analyzing resected tumor tissues. To the best of our knowledge, no other study have compared qMRI parameters measured in vivo to ex vivo iron histology and gene expression of the same human tissue. Taken together, the different validation strategies all indicate that the $r_1$-$r_2^*$ relaxivity increases the specificity of MRI to different molecular environments of iron, highlighting the robustness of our findings.

Our proposed approach for measuring iron homeostasis in vivo using the $r_1$-$r_2^*$ relaxivity may have wide clinical and scientific implications. First, the $r_1$-$r_2^*$ relaxivity provides a unique contrast for imaging the brain, which is associated with 45 distinct gene sets not associated with $R_1$ or $R_2^*$ by themselves. Moreover, we show that the $r_1$-$r_2^*$ relaxivity, which captures paramagnetic properties, enhances the contrast between tumor tissue and normal-appearing white-matter and gray-matter tissues. In agreement with these findings, meningioma tumors have been shown to contain a higher concentration of ferrimagnetic particles and an abnormal expression of iron-related genes compared to non-pathological brain tissue[17,18]. Indeed, the contrast enhancement we saw with the $r_1$-$r_2^*$ relaxivity is similar to the one observed for Gd-enhanced imaging, which is based on the altered relaxivity in the presence of paramagnetic agents[76]. Concerns regarding the safety of Gd-based contrast agents raise the need for Gd-free diagnosis of brain tumors[56,57]. Adjusting our approach for clinical imaging might offer safer alternatives for brain tumor diagnosis.

Finally, the sensitivity of the $r_1$-$r_2^*$ relaxivity to iron homeostasis in the brain may have clinical implications for neurodegenerative diseases. Alterations in the distribution of molecular iron compounds can lead to cellular damage which is disease-specific[10]. In particular, the iron mobilization capacity was shown to differ between elderly controls and patients diagnosed with either Parkinson's disease (PD) or Alzheimer's disease (AD)[7]. We found that this iron homeostasis marker is correlated with the $r_1$-$r_2^*$ relaxivity across brain regions and age groups. This result, in addition to our other validation strategies, demonstrates the sensitivity of the $r_1$-$r_2^*$ relaxivity to the iron homeostasis in the brain and in the aging process. Therefore, our approach can add an important layer of information to existing in vivo PD and AD biomarkers such as neuromelanin MRI[12].

To conclude, we present an MRI contrast, based on the $r_1$-$r_2^*$ relaxivity, which is sensitive to the iron homeostasis in the human brain. This technology can differentiate between tumor tissue and non-pathological tissue without injecting contrast agents, and can detect biological properties inaccessible to conventional MRI approaches. We validated the sensitivity of the $r_1$-$r_2^*$ relaxivity to the molecular state of iron using in vitro, in vivo and ex vivo analyses. We show that our MRI technology reveals the intrinsic paramagnetic properties of different in vitro iron environments. Furthermore, the $r_1$-$r_2^*$ relaxivity varies with the iron mobilization capacity across brain regions and age groups, and reveals differences in iron homeostasis and iron-related gene expression in pathological tissues. Therefore, this approach may further advance our understanding of the impaired iron homeostasis in cancer, normal aging and neurodegenerative diseases, and may open new avenues for the non-invasive research and diagnosis of the living human brain.

## Methods

### In vivo iron relaxivity model

The iron relaxivity model assumes a linear relationship between relaxation rates and iron concentration (Eq. (1))[30–33].

This linear relationship for two different iron environments $a$ and $b$ with iron concentrations [a] and [b] can be expressed using the following equations:

$$R_1 = r_{(1,a)}[a] + c_{(1,a)} \qquad R_1 = r_{(1,b)}[b] + c_{(1,b)} \tag{3}$$

$$R_2^* = r_{(2,a)}[a] + c_{(2,a)} \qquad R_2^* = r_{(2,b)}[b] + c_{(2,b)} \tag{4}$$

where $r_{(1/2,a/b)}$ represents the $R_1$-iron relaxivity or the $R_2^*$-iron relaxivity of the two iron environments, and $c_{(1/2,a/b)}$ are constants.

The two iron environments are distinguished by their iron relaxivities under the assumption:

$$r_{(1,a)} \neq r_{(1,b)} \qquad and/or \qquad r_{(2,a)} \neq r_{(2,b)} \tag{5}$$

Rearranging Eq. (4):

$$[a] = \frac{R_2^* - c_{(2,a)}}{r_{(2,a)}} \qquad [b] = \frac{R_2^* - c_{(2,b)}}{r_{(2,b)}} \tag{6}$$

Substituting Eq. (6) in Eq. (3):

$$R_1 = \frac{r_{(1,a)}}{r_{(2,a)}} R_2^* + c_a \qquad R_1 = \frac{r_{(1,b)}}{r_{(2,b)}} R_2^* + c_b \tag{7}$$

Where $\frac{r_{(1,a)}}{r_{(2,a)}}$ and $\frac{r_{(1,b)}}{r_{(2,b)}}$ represent the linear dependencies of $R_1$ on $R_2^*$ ($r_1$-$r_2^*$ relaxivities) of the two iron environments $a$ and $b$ (see Eq. (2)). The intercept of the linear dependency of $R_1$ on $R_2^*$ can be described by the following expression: $c_{a/b} = c_{(1,a/b)} - \frac{r_{(1,a/b)}}{r_{(2,a/b)}} c_{(2,a/b)}$. For a discussion on

the biophysical meaning of the $R_1$-$R_2^*$ intercept see Supplementary Section 7.

Importantly, the MRI-measured $r_1$-$r_2^*$ relaxivity serves as an in vivo estimator of iron relaxivity and reveals intrinsic properties of the iron environment.

Assuming that the iron relaxivity of $R_1$ provides a different separation between the two iron environments $a$ and $b$ compared to the iron relaxivity of $R_2^*$:

$$\frac{r_{(1,a)}}{r_{(2,a)}} \neq \frac{r_{(1,b)}}{r_{(2,b)}}$$

These two iron environments $a$ and $b$ then can be distinguished by their $r_1$-$r_2^*$ relaxivities (i.e., the in vivo iron relaxivities). In the brain, the $r_1$-$r_2^*$ relaxivity could be affected by the entire milieu of available iron compounds, their iron binding capacities and aggregation states (i.e. the molecular iron environment). For an elaboration of this model under the assumption of a complex iron environment and in the presence of myelin see Supplementary Section 1.

### Phantom samples experiments
**Phantom system.** We prepared samples of four different iron compounds: transferrin (holo-transferrin human, Sigma), apo-transferrin (apo-transferrin human, Sigma), ferritin (equine spleen, Sigma), and ferrous (iron (II) sulfate heptahydrate, Sigma). These samples were prepared in three different molecular environments: liposomes, 18.2 MΩcm water, and bovine serum albumin (BSA, Sigma)[77,78]. For each combination of iron compound and molecular environment, we made different samples by varying both the iron compound concentration and the lipid/BSA-water fractions. For liposomal/BSA environments, the iron compounds concentrations were divided by the water-lipid or water-BSA fractions to get units of [mg/wet ml]. The liposomes were made from a mixture of soy phosphatidylcholine (PC) and egg sphingomyelin (SM) purchased from Lipoid and used without further purification. Additional results with PC and PC-cholesterol (Sigma) liposomes are presented in Supplementary Fig. 12. The lipid samples were mixed in chloroform at desired mole ratios and evaporated under reduced pressure (8 mbar) in a Buchi rotary evaporator vacuum system (Flawil, Switzerland). The resulting lipid film was resuspended in a 10 mM ammonium bicarbonate solution, lyophilized, and subsequently hydrated in the reassembly buffer. To achieve the desired lipid-protein concentration, the protein solution (~50 mg/ml in water) was diluted to the right concentration and subsequently was added to the lyophilized lipid powder. For the BSA phantoms, samples were prepared by dissolving lyophilized BSA in 18.2 MΩcm water at the desired concentrations. For the Hemoglobin phantoms, samples were prepared by dissolving lyophilized human hemoglobin powder (Sigma) in 18.2 MΩcm water at the desired concentrations. The hemoglobin concentrations in this experiment were constrained by its solubility (20 mg/mL for hemoglobin alone, and lower when it is in a mixture with other iron-binding proteins).

Samples were placed in a 2-ml glass vials glued to a glass box, which was then filled with ~1% SeaKem® LE Agarose (Ornat) and 0.1% gadolinium (Gadoteric acid (Dotarem, Guerbet)) dissolved in distilled water (DW). The purpose of the agarose with gadolinium (Agar-Gd) was to stabilize the vials and to create a smooth area in the space surrounding the samples that minimized air-sample interfaces.

### MRI acquisition for phantoms
Data were collected on a 3T Siemens MAGNETOM Skyra scanner equipped with a 32-channel head receive-only coil at the ELSC Neuroimaging Unit at the Hebrew University.

**Quantitative $R_1$ and MTV**. 3D Spoiled gradient echo (SPGR) images were acquired with different flip angles (FA = 4°, 8°, 16°, and 30°). The

TE/TR were 4.45/18 ms. The scan resolution was 0.5 mm × 0.5 mm × 0.6 mm. For B1+ mapping, we acquired an additional spin-echo inversion recovery scan (SEIR). This scan was done on a single slice, with an adiabatic inversion pulse and inversion times of TI = 2,000, 1,200, 800, 400, and 50 ms. The TE/TR were 73/2,540 ms. The scan resolution was 1.2 mm × 1.2 mm × 2.0 mm.

**Quantitative $R_2^*$.** SPGR images were acquired with different flip angles (α = 4°, 8°, 16°, and 30°). The TR was 27 ms and 5 echoes were equally spaced between 4.45 and 20.85 ms. The scan resolution was 0.5 mm × 0.5 mm × 0.6 mm.

**Quantitative $R_2$.** Multi spin-echo images with 15 equally spaced spin echoes between 10.5 ms and 157.5 ms. The TR was 4.94 s. The scan resolution was 1.2 mm isotropic.

### Estimation of qMRI parameters for phantoms
**Quantitative $R_1$ and MTV mapping.** $R_1$ and MTV estimations for the lipid samples were computed with the mrQ[46] (https://github.com/mezera/mrQ) and Vista Lab (https://github.com/vistalab/vistasoft/wiki) software packages. The mrQ software was modified to suit the phantom system[78]. The modification utilizes the fact that the Agar-Gd mixture which fills the box around the vials is homogeneous, and therefore can be assumed to have a constant $R_1$ value. We used this gold-standard $R_1$ value generated from the SEIR scans to correct for the inhomogeneous excitation bias in the SPGR scans.

A mask labeling the different phantom samples was generated based on MATLAB's "imfindcircles" function, and was filtered to remove voxels with extreme signals. Voxels were filtered based on a fixed threshold on the SPGR signal at FA = 16. In addition, we also filtered out those voxels in which the SPGR signal at FA = 16 was two median absolute deviations away from the median value. We further edited this mask manually, removing voxels with susceptibility artifacts resulting from the vials and air pockets. To fit the $R_1$ and proton density of each phantom sample, we calculated the median values of the SPGR signal as well as the inhomogeneous excite and receive biases across all the voxels of each sample. These median values were used in the Vista Lab function "relaxFitT1" to find the median $R_1$ and proton density of each sample. proton density values then were calibrated using the proton density of a water-filled vial in order to calculate the MTV values.

**Quantitative $R_2^*$ mapping.** We used the SPGR scans with multiple echoes to estimate $R_2^*$. Fitting was done by taking the median values of the SPGR signal across all the voxels of the phantom sample for each TE. To label the different samples, we used the same mask that was used to calculate $R_1$ and MTV. We then used an exponential fitting process to find $R_2^*$. As we had four SPGR scans with variable flip angles, we averaged the $R_2^*$ values acquired from each of these scans for increased signal to noise ratio.

**Quantitative R2 mapping.** We used spin echo scans with multiple echoes to estimate R2. Fitting was done by taking the median values of the spin-echo signal across all the voxels of the phantom sample for each TE. To label the different samples, we used the same mask that was used to calculate $R_1$ and MTV. We then used an exponential fitting process to find R2.

### $r_1$-$r_2^*$ relaxivity computation for phantoms
For each iron compound in each molecular environment, we computed the linear dependency of $R_1$ on $R_2^*$ across samples with varying iron-binding protein concentrations relative to the water fractions. We fitted the following linear model across samples:

$$R_1 = a \cdot R_2^* + b \tag{8}$$

The slope of this linear model ($a$) represents the $r_1$-$r_2^*$ relaxivity Eq. (2) and (7). $b$ is the intercept (the residual $R_1$ not explained by $R_2^*$). This process was implemented in MATLAB.

## Estimation of total iron content in phantoms

We estimated the iron content of our transferrin and ferritin samples using the following equation:

$$\text{iron}\left(\frac{\text{mg}}{\text{ml}}\right) = \frac{\text{iron binding protein}\left(\frac{\text{mg}}{\text{ml}}\right)}{\text{protein molecular weight}\left(\frac{\text{mg}}{\text{mol}}\right)} \cdot \frac{\text{iron ions}}{\text{protein}} \cdot \text{iron molecular weight}\left(\frac{\text{mg}}{\text{mol}}\right)$$

(9)

Transferrin contains 2 iron ions per protein molecule[3], and its molecular weight was estimated as $76 \times 10^6$ mg/mol (based on manufacturer information). The iron loading of ferritin was estimated as 2,250 iron ions per protein molecule (based on manufacturer information) and its molecular weight was estimated as $440 \times 10^6$ mg/mol[3]. The molecular weight of iron was set to $55.847 \times 10^3$ mg/mol.

This resulted in the following equation for converting iron-binding protein concentrations into iron concentrations:

1 mg/ml transferrin = 1.4 μg/ml iron

1 mg/ml ferritin = 0.29 mg/ml iron

Importantly, this model is only applicable in vitro, for controlled and homogeneous iron environments. We do not apply this molecular model to MRI values of brain tissue which is complex and heterogenous.

## Human MRI datasets

Table 1 summarizes the different human MRI datasets used in the study.

## Healthy human subjects

We scanned 26 young adults (aged $27 \pm 10$ years, 10 females), and 13 older adults (aged $70 \pm 3$ years, 4 females). Healthy volunteers were recruited from the community surrounding the Hebrew University of Jerusalem. The experimental procedure was approved by the Helsinki Ethics Committee of Hadassah Hospital, Jerusalem, Israel. Written informed consent was obtained from each participant prior to the procedure. This data was first used in our previous work[77]. Sex and gender were determined based on self-report were not considered in the study design as this information is irrelevant for our findings.

## Meningioma patients

During the study period May 2019 to August 2020, we recruited 19 patients who had undergone surgery for the resection of brain meningiomas. All patients had preoperative qMRI scans in addition to their clinical brain MRI assessment. One subject, with a titanium cranial fixation plate adjacent to the tumor, was excluded from the study due to local disruption of the magnetic field. The final cohort included 18 patients (11 females). Imaging studies were anonymized before they were transferred for further analysis. Brain meningioma surgical specimens, available for 16 patients, were obtained from the fresh frozen tissue biobank of the Department of Neurosurgery, Shaare Zedek Medical Center, Jerusalem, Israel, and were transferred on dry ice for western-blot and gene expression analyses. The experimental procedure was approved by the Helsinki Ethics Committee of Shaare Zedek Medical Center, Jerusalem, Israel. Study participants provided informed consent according to an institutional review board, approval numbers 0299-17 SZMC and 0159-20 SZMC. Sex and gender were determined based on self-report were not considered in the study design as this information is irrelevant for our findings.

## MRI acquisition for healthy human subjects

Data were collected on a 3T Siemens MAGNETOM Skyra scanner equipped with a 32-channel head receive-only coil at the ELSC Neuroimaging Unit at the Hebrew University.

**Quantitative $R_1$, $R_2^*$ and MTV mapping.** SPGR echo images were acquired with different flip angles ($\alpha = 4°$, $10°$, $20°$, and $30°$). Each image included 5 equally spaced echoes (TE = 3.34–14.02 ms) and the TR was 19 ms. The scan resolution was 1 mm isotropic. Additional SPGR echo image was acquired with an MT pulse (TE = 3.34, TR = 27, $\alpha = 10$, 1 mm isotropic). For B1+ mapping, we acquired an additional spin-echo inversion recovery scan with an echo-planar imaging read-out (SEIR-epi). This scan was done with a slab-inversion pulse and spatial-spectral fat suppression. For SEIR-epi, the TE/TR were 49/2,920 ms. The TIs were 200, 400, 1200, and 2400 ms. We used 2-mm in-plane resolution with a slice thickness of 3 mm. The EPI readout was performed using 2× acceleration.

**Anatomical images.** 3D magnetization-prepared rapid gradient echo (MPRAGE) scans were acquired for 30 of the 39 healthy subjects. The scan resolution was 1 mm isotropic, the TE/TR were 2.98/2300 ms. Magnetization-prepared 2 rapid acquisition gradient echo (MP2RAGE) scans were acquired for the remaining 9 subjects. The scan resolution was 1 mm isotropic, the TE/TR were 2.98/5000 ms.

**Whole-brain DTI measurements.** performed using a diffusion-weighted spin-echo EPI sequence with isotropic 1.5-mm resolution. Diffusion weighting gradients were applied at 64 directions and the strength of the diffusion weighting was set to b = 2000s/mm² (TE/TR = 95.80/6000 ms, $G$ = 45mT/m, δ = 32.25 ms, Δ=52.02 ms). The data includes eight non-diffusion-weighted images ($b = 0$). In addition, we

## Table 1 | human MRI datasets used in the study

| Subjects | MRI protocol |
|---|---|
| Healthy Human subjects (3 T Siemens MAGNETOM Skyra): 26 young adults (aged $27 \pm 10$ years, 10 females), and 13 older adults (aged $70 \pm 3$ years, 4 females) | • SPGR ($\alpha = 4°$, $10°$, $20°$ and $30°$, TE = 3.34–14.02 ms, TR = 19 ms, 1X1X1 mm, bandwidth=430 Hz/Px, 2 × acceleration)<br>• SEIR-epi (TE/TR = 49/2,920 ms, TI = 200, 400, 1,200, and 2,400 ms, 2X2X3 mm, bandwidth=1776 Hz/Px, 2 × acceleration)<br>• SPGR with an MT pulse (TE/TR = 3.34/27 ms, $\alpha = 10°$, 1X1X1 mm, bandwidth=430 Hz/Px, 2 × acceleration)<br>• MPRAGE (TE/TR = 2.98/2,300 ms, 1X1X1 mm) or MP2RAGE (TE/TR = 2.98/5,000 ms, 1X1X1 mm, bandwidth=240 Hz/Px, 2 × acceleration)<br>• Diffusion-weighted spin-echo EPI (isotropic 1.5-mm, 64 directions; b = 2000s/mm² + 8 directions b = 0 s/mm², TE/TR = 95.80/6,000 ms, G = 45mT/m, δ = 32.25 ms, Δ = 52.02 ms, bandwidth = 1450 Hz/Px, 2 × acceleration) + non-diffusion-weighted images with reversed phase-encode blips (P»A). |
| Meningioma patients (3T Siemens MAGNETOM Skyra):18 patients (11 females) | • SPGR ($\alpha = 4°$, $10°$, $20°$ and $30°$, TE = 2.85–14.02 ms, TR = 18 ms, 1.5 × 1.5 × 1.5 mm, bandwidth = 425 Hz/Px, 2 × acceleration)<br>• SEIR-epi (TE/TR = 49/2,920 ms, TI = 200, 400, 1,200, and 2,400 ms, 2X2X3 mm, bandwidth=1775 Hz/Px, 2 × acceleration)<br>• Gd-enhanced MPRAGE (TE/TR = 2.4/1,800 ms, 1X1X1 mm). |

collected non-diffusion-weighted images with reversed phase-encode blips. For two subjects (1 young, 1 old) we failed to acquire this correction data and they were excluded from the diffusion analysis.

### MRI acquisition for meningioma patients

Data were collected on a 3T Siemens MAGNETOM Skyra scanner equipped with a 32-channel head receive-only coil at the Shaare Zedek Medical Center.

**Quantitative $R_1$, $R_2^*$ and MTV mapping.** SPGR echo images were acquired with different flip angles ($\alpha = 4°$, 10°, 20°, and 30°). Each image included 5 equally spaced echoes (TE = 2.85–14.02 ms) and the TR was 18 ms. The scan resolution was 1.5 mm isotropic. For calibration, we acquired an additional SEIR-epi scan. This scan was done with a slab-inversion pulse and spatial-spectral fat suppression. For SEIR-epi, the TE/TR were 49/2,920 ms. The TIs were 200, 400, 1200, and 2400 ms. We used 2-mm in-plane resolution with a slice thickness of 3 mm. The EPI readout was performed using 2× acceleration.

**Gd-enhanced anatomical images.** Gd-enhanced MPRAGE scans were acquired. The scan resolution was 1 mm isotropic, the TE/TR were 2.4/1800 ms. The contrast agent was either Multihance or Dotarem at a dose of 0.1 mmol/kg. Contrast agent injection and MPRAGE acquisition were done after the acquisition of the quantitative MRI protocol, or on a different day.

### Estimation of qMRI parameters for human subjects

**Quantitative $R_1$ and MTV mapping.** Whole-brain MTV and $R_1$ maps, together with bias correction maps of B1+ and B1-, were computed using the mrQ software[46,79].

**Quantitative $R_2^*$ mapping.** We used the SPGR scans with multiple echoes to estimate $R_2^*$. Fitting was performed with the Voxel Based Quantification (VBQ) toolbox (v2e) in SPM12[80]. As we had four SPGR scans with variable flip angles, we averaged the $R_2^*$ maps acquired from each of these scans for increased SNR.

**Quantitative MD mapping.** Diffusion analysis was done using the FDT toolbox in FSL[81,82]. Susceptibility and eddy current induced distortions were corrected using the reverse phase-encode data, with the eddy and topup commands[83,84]. MD maps were calculated using vistasoft (https://github.com/vistalab/vistasoft/wiki).

**Quantitative MTsat mapping.** MTsat maps were computed as described in Helms et al. (2008)[47]. The MTsat measurement was extracted from the equation:

$$MTsat = M_0(B1+) \cdot \alpha \frac{R1\,TR}{S_{MT}} - \frac{((B1+) \cdot \alpha)^2}{2} - R1\,TR \qquad (10)$$

Where $S_{MT}$ is the signal of the SPGR scan with additional MT pulse, $\alpha$ is the flip angle and TR is the repetition time. $M_0$ (the equilibrium magnetization parameter), $B1+$ (the transmit inhomogeneity) and $R1$ estimations were computed from the non-MT weighted SPGR scans, during the pipeline described under "*Quantitative $R_1$ & MTV mapping*". Registration of the $S_{MT}$ image to the imaging space of the $R_1$ map was done using a rigid-body alignment ($R_1$, B1+ and $M_0$ are all in the same space).

### Brain segmentation in healthy subjects

Whole-brain segmentation was computed automatically using the FreeSurfer segmentation algorithm (v6.0)[85]. For subjects with MPRAGE scan, we used that as a reference; for the other subjects the MP2RAGE scan was used. These anatomical images were registered to the $R_1$ space prior to the segmentation process, using a rigid-body alignment.

FreeSurfer's estimates of subcortical gray-matter structures were replaced with estimates from FSL's FIRST tool[86].

### Brain segmentation in meningioma patients

Tumor contouring was performed by the neurosurgeon (T.S.) using BrainLab's Elements software, Cranial Navigation 3.1.5 (BrainLab AG, Munich, Germany) over the Gd-enhanced MPRAGE images, and exported as a DICOM file for further analysis. The contours of the tumors were registered to the $R_1$ space using rigid-body segmentation. Cases that required manual adjustment were examined and approved for accuracy by the neurosurgeon (T.S.).

Whole-brain segmentation was computed automatically using the FreeSurfer segmentation algorithm (v6.0)[85]. We used the synthetic T1w image generated with mrQ as the reference image, from which we removed the skull and the tumor. We then ran FreeSurfer with the "-noskullstrip" flag. For each patient, we used FreeSurfer's segmentation in the tumor-free hemisphere. Estimates for the entire white-matter and gray-matter tissues were averaged across the different FreeSurfer parcellations in these regions.

### $r_1$-$r_2^*$ relaxivity computation for ROIs in the human brain

We used MATLAB to compute the $r_1$-$r_2^*$ relaxivity in different brain areas. For each ROI, we extracted the $R_2^*$ and $R_1$ values from all voxels. $R_2^*$ values were pooled into 36 bins spaced equally between 0 and $50\ s^{-1}$. This was done so that the linear fit would not be heavily affected by the density of the voxels in different $R_2^*$ regimes. We removed any bins in which the number of voxels was smaller than 4% of the total voxel count in the ROI. The median $R_2^*$ of each bin was computed, along with the $R_1$ median. We used these data points to fit the following linear model across bins using simple linear regression Eq. (8).

$$R1 = a \cdot R_2^* + b \qquad (11)$$

The slope of this linear model ($a$) represents the $r_1$-$r_2^*$ relaxivity. $b$ is the intercept (the residual $R_1$ not explained by $R_2^*$).

### Generating voxel-wise $r_1$-$r_2^*$ relaxivity visualizations

In order to generate a voxel-wise visualization of the $r_1$-$r_2^*$ relaxivity in the brain, we calculated the local linear dependency of $R_1$ on $R_2^*$ using a moving-window approach. For each voxel within the brain mask, we extracted $R_1$ and $R_2^*$ values of that voxel and all its neighboring voxels (a box of 125 voxels total). If at least 10 of these voxels were inside the brain mask, we fit the following linear model across these voxels Eq. (8).

$$R1 = a \cdot R_2^* + b \qquad (12)$$

The slope of this linear model ($a$) represents the local $r_1$-$r_2^*$ relaxivity of the voxel. $b$ is constant.

### Group-level comparison of qMRI parameters and histological measurements

We aggregated data published in different papers[5,7,9] that describe ferritin, transferrin and iron concentrations in 11 brain regions. One of the papers[9] described the concentration of L-rich ferritin and H-rich ferritin independently and we combined these estimates for each ROI to get the total ferritin concentration. One of the papers[5] reported the iron level in units of [μmol Fe/ g protein] and we converted these measurements to units of [μg Fe/g protein]. In order to use this data for our analysis we matched the brain regions reported in the literature with their corresponding FreeSurfer labels. Table 2 summarizes the aggregated data.

**Table 2 | Iron compounds concentrations reported in the literature for different brain regions (with references and sample sizes) and the corresponding FreeSurfer labels**

| Brain Region | FreeSurfer labels[a] | Age group | Transferrin [ng/µg protein][b] | Ferritin [ng/µg protein][c] | Iron [mg/g protein][d] |
|---|---|---|---|---|---|
| Frontal CTX | 1003,1012,1014,1019,1020,1027,1028,1032 | younger | 3.88[7] (N = 9) | 45.34[9] (N = 9) | 1.21[7] (N = 9) |
| | | older | 4.09[7] (N = 11) | 89.77[9] (N = 11) | 1.59[7] (N = 11) |
| Caudate | 11 | younger | 3.21[7] (N = 8) | 60.92[9] (N = 7) | 1.79[7] (N = 8) |
| | | older | 4.44[7] (N = 8) | 135.12[9] (N = 8) | 3.34[7] (N = 8) |
| Putamen | 12 | younger | 3.49[7] (N = 8) | 58.63[9] (N = 7) | 2.59[7] (N = 8) |
| | | older | 4.47[7] (N = 8) | 124.12[9] (N = 8) | 4.62[7] (N = 8) |
| Substantia nigra | 173 | younger | 2.45[7] (N = 8) | 54.96[9] (N = 7) | 2.92[7] (N = 8) |
| (midbrain)[e] | | older | 3.42[7] (N = 8) | 135.12[9] (N = 8) | 5.70[7] (N = 8) |
| Globus pallidus[f] | 13 | younger | 4.82[7] (N = 8) | 28.40[9] (N = 7) | 7.39[7] (N = 8) |
| | | older | 5.10[7] (N = 8) | 215.27[9] (N = 8) | 4.07[7] (N = 8) |
| Gray superior temporal gyrus | 1001,1006,1007,1009, 1015,1016,1030,1033,1034 | younger | 1.84[5] (N = 9) | 25.65[5] (N = 9) | 0.45[5] (N = 9) |
| (Temporal CTX)[e] | | older | 1.08[5] (N = 11) | 24.20[5] (N = 11) | 0.69[5] (N = 11) |
| White superior temporal gyrus | 3001,3006,3007,3009, 3015,3016,3034,3030,3033 | younger | 3.82[5] (N = 9) | 44.87[5] (N = 9) | 0.72[5] (N = 9) |
| (Temporal WM)[e] | | older | 2.52[5] (N = 11) | 33.50[5] (N = 11) | 0.74[5] (N = 11) |
| Motor CTX | 1017,1022,1024,1031 | younger | 2.91[5] (N = 9) | 40.20[5] (N = 9) | 1.16[5] (N = 9) |
| | | older | 2.93[5] (N = 11) | 22.13[5] (N = 11) | 0.89[5] (N = 11) |
| Motor WM | 3024,3017,3022,3031 | younger | 5.75[5] (N = 9) | 30.38[5] (N = 9) | 1.39[5] (N = 9) |
| | | older | 7.34[5] (N = 11) | 21.51[5] (N = 11) | 1.25[5] (N = 11) |
| Occipital CTX | 1011,1013,1005,1021 | younger | 1.75[5] (N = 9) | 9.34[5] (N = 9) | 0.50[5] (N = 9) |
| | | older | 1.67[5] (N = 11) | 34.27[5] (N = 11) | 1.16[5] (N = 11) |
| Occipital WM | 3011,3013,3005,3021 | younger | 3.53[5] (N = 9) | 13.81[5] (N = 9) | 0.80[5] (N = 9) |
| | | older | 6.07[5] (N = 11) | 31.38[5] (N = 11) | 0.90[5] (N = 11) |

*WM* white matter, *CTX* cortex.

[a]These labels represent left-hemisphere ROIs, but the corresponding right-hemisphere labels were used as well. For each subject, we averaged the MRI measurements of both hemispheres of bilateral brain regions.

[b]Transferrin levels were determined by ELISA[7], or by SDS-PAGE and immunoassay with western blotting[5]. In both works transferrin levels were adjusted for total protein as determined with the Bio-Rad Protein Assay (Bio-Rad)[5, 7].

[c]Ferritin levels were determined by immunoassays with slot blot technique, and were adjusted for total protein as determined with the Bio-Rad Protein Assay (Bio-Rad) in both works[5, 9].

[d]Iron levels were determined by Ferrochem II Serum Iron / TIBC analyzer, and were adjusted for total protein as determined with the Bio-Rad Protein Assay (Bio-Rad)[5, 7].

[e]To avoid very small and unreliable ROI segmentations, for the substantia nigra we used the entire midbrain, and for the gray/white superior temporal gyrus we used the entire temporal CTX/WM.

[f]For results when the pallidum is excluded see Supplementary Section 12.

## Western blot analysis of meningioma tissue

Fresh frozen meningioma samples (40–50 mg) from 16 patients were homogenized in 200 µL of RIPA buffer (Thermo Fisher Scientific) supplemented with protease inhibitor (Sigma-Aldrich) using a Bioruptor Pico sonication device (Diagenode) and protein extraction beads (Diagenode, NJ, USA) according to the manufacturer instructions. Protein concentration was determined using the Pierce assay (Thermo Fisher Scientific, MA, USA). Samples containing 20 µg of protein were separated on 4–20% Tris-Glycine SDS-PAGE gel (Bio-Rad, CA, USA) and transferred to PVDF membrane using Trans-Blot Turbo transfer system and transfer packs (Bio-Rad, Hercules, CA, USA). Membranes were probed using Anti-Ferritin Light chain (#AB69090, Abcam, 1:1,000 dilution) and Anti-Transferrin (#AB82411, Abcam, 1:10,000 dilution) primary antibodies and appropriate horseradish peroxidase-conjugated secondary antibody (#AB6721, Abcam, 1:20000 dilution). Membranes were treated with EZ-ECL (Biological industries, Beit-Ha'emek, Israel) and visualized using ImageQuant LAS 4000 (GE Healthcare, IL, USA). Blot intensities were quantified using the FIJI ImageJ software (v2.35)[87]. The ratio of transferrin/ferritin was based on the ratio in the blot intensities of transferrin and ferritin. Due to the noisy nature of the western-blot analysis, we averaged the estimates over six repetitions. We then used the median transferrin/ferritin ratio across subjects (which was equal to 1) as the threshold between the two groups (low and high transferrin/ferritin ratio).

## RNA-sequencing of meningioma tissue

**RNA-seq libraries.** Tumor samples from 17 patients (samples from 16 patients and a replicate for one) were flash frozen and kept in −80c until processing. RNA isolation was done with the following steps: First, frozen tissue was chopped and transferred with a 2 ml lysis buffer (Macherey-Nagel, 740955) five times through a needle attached to a 0.9 mm syringe to achieve homogenization. Next, total RNA was extracted with NucleoSpin RNA kit (Macherey-Nagel, 740955), following the standard protocol. Finally, mRNA was isolated using the NEBNext Poly(A) mRNA Magnetic Isolation Module (NEB E7490S), using 5µg of total RNA as an input and following the standard protocol. The purified mRNA was used as input for cDNA library preparation, using NEBNext® Ultra™ II Directional RNA Library Prep Kit for Illumina (NEB E7760), and following the standard protocol. Quantification of the libraries was done by Qubit and TapeStation. Paired-end sequencing of the libraries was performed on Nextseq 550.

**Data processing.** The demultiplexing of the samples was done with Illumina's bcl2fastq software (v2.19.1.403). The fastq files were next aligned to the human genome (human genome assembly GRCh38) using STAR (v2.7.3a) and the transcriptome alignment and gene counts were obtained with HTseq (v0.13.5). For quality control RNAseQC software was used (Picard v2.26.10). Quality control and data normalization were done in R using the DEseq2 package from

Bioconductor (v3.13). The counts matrix per gene and sample were normalized using the Variance stabilizing transformation. Genes with less than 5 counts were filtered out of the analysis. The filtered and normalized matrix was used in all downstream analysis.

### Gene set enrichment analysis (GSEA)

The final sequencing dataset included the expression of approximately 27,000 genes in 17 tumor samples. We then excluded unannotated genes based on the gene ontology resource (http://geneontology.org/) as well as genes with low (<6) expression levels, yielding 19,500 genes.

We used GSEA software (v4.1.0) to further validate that the subset of highly correlated genes is not random, but rather represents known biological pathways. For this aim, we calculated the correlations across patients between the expression of each of the genes and one of the qMRI parameters ($R_1$, $R_2^*$ or $r_1$-$r_2^*$ relaxivity). For each of the qMRI parameters, genes were ranked based on the $r$ values of the correlations, and the ranked list was used in the GSEAPranked toolbox[59,60]. The gene sets databases used for this analysis included go, biocarta, kegg, pid, reactome and wikipathways.

The primary result of the GSEA is the enrichment score (ES), which reflects the degree to which a gene set is overrepresented at the top or bottom of a ranked list of genes: a positive ES indicates gene set enrichment at the top of the list, while a negative ES indicates gene set enrichment at the bottom. The normalized ES (NES) accounts for differences in gene set size and in correlations between gene sets and the expression dataset.

One tumor sample was excluded from the analysis, as the $R_1$ and $R_2^*$ values in the tumor were relatively high, which led to the fact that no significantly enriched pathway were found for $R_1$ and $R_2^*$ (though we did detect significantly enriched pathways for the $r_1$-$r_2^*$ relaxivity). Removing this outlier improved GSEA results for $R_1$ and $R_2^*$ and we therefore excluded this subject.

Following the GSEA analysis, we found a total of 101 significantly enriched pathways for at least one of the $R_1$, $R_2^*$ and $r_1$-$r_2^*$ relaxivity. We then clustered those significantly enriched pathways using the "clustergram" function in MATLAB. In order to evaluate which genes are included within the top enrichment pathways for each MRI parameter, we used Leading Edge Analysis (as implemented in the GSEA toolbox).

### Reporting summary

Further information on research design is available in the Nature Portfolio Reporting Summary linked to this article.

## Data availability

The raw and processed RNA-sequencing data generated in this study have been deposited in the GEO database under accession code GSE240204. MRI and proteomics measurements for each Meningioma patient, and MRI measurements for healthy subjects at all presented ROIs are available in the source data file. The raw data are not publicly available due to them containing information that could compromise research participant privacy/consent. Anonymized images and any additional raw data are available from the corresponding author [S.F.] upon request which conform with the privacy guidelines of the Hadassah Hospital and Shaare Zedek Medical Center Helsinki Ethics Committees. The timeframe for response to such requests is within a month. Source data are provided with this paper.

## Code availability

A toolbox for computing the $r_1$-$r_2^*$ relaxivity[88], including example data, is available at: https://doi.org/10.5281/zenodo.8186069.

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

## Acknowledgements
This work was supported by the ISF grant 1169/20, awarded to A.A.M. S.F is grateful to the Azrieli Foundation for the award of an Azrieli Fellowship. This work is dedicated to the memory of Mrs. Lily Safra, a great supporter of brain research.

## Author contributions
S.F. and A.A.M. conceived of the presented idea. S.F. wrote the manuscript and designed the figures. A.A.M, R.S., N.H. and T.S. commented and edited the manuscript. S.F. collected the healthy human subjects datasets and analyzed the healthy human subjects, meningioma patients, phantom and gene-expression datasets. R.S. conceived, designed and performed the phantom experiments and wrote the relevant methods section. M.A., D.B.H and R.S. performed the RNA-sequencing. N.H. assisted and instructed with the RNA sequencing and gene-expression analysis. E.B.D, N.M. and T.S. collected the meningioma patients datasets. H.S. and T.S. performed the western-blot analysis of meningioma tissue.

## Competing interests
A.A.M. and T.S. research groups are partly funded by Integra Holdings. This funding was used for the collection of some of the MRI data, for other purposes unrelated to the submitted manuscript. The funders do not have any role in the conceptualization, design, analysis, decision to publish, or preparation of the manuscript. None of the other authors have any financial or non-financial conflicts of interest.
