## [Peer Review File · Nature Communications]

Non-invasive assessment of normal and impaired iron homeostasis in the brainReviewer #1 (Remarks to the Author):

In this new manuscript, Filo et al use a relation between R1 and R2* relativity, which are MRI parameters sensitive to iron, to estimate tissue concentrations in brain of transferrin and ferritin – the major non-heme binding proteins. In so doing, they present a new contrast that may have utility in different settings. The approach they investigate is brain cancer, where they show the advantage over other techniques. The authors begin by characterising the signals for transferrin and ferritin in vitro, before applying this to human brains, and referencing these changes to transcriptional profiles and historical measures of brain iron. These are novel and important findings that may yet be further developed to investigate other iron binding partners (heme?) and applied in different contexts. While there will inevitably be questions (that may not be answerable) as to the validity of this measure for certain species of iron, the authors have developed their argument well, and they have introduced a new contrast for the field regardless.

I have the following questions:

1. both R1 and R2* have the same unit. why not perform a voxel-wise division? The current approach where each ROI is binned followed by a regression fitting seems to smooth the data and gather all the important contributions of other sources in the constant. It would also be interesting to look at the intercept of the fit.
2. The r1-r2* relaxivity approach assumes the R1 and R2* signal is only coming from non-heme (ferritin and Transferrin) paramagnetic sources (or the extended version from Myelin). In in vivo tissues, there is a large contribution from iron from hemoglobin. The current resolution of imaging may not allow to capture this. Could an in vitro experiment demonstrate that r1-r2* relaxivity of ferritin or transferrin is not impacted by hemoglobin when it is added? – possibly at 10 fold greater concentrations to reflect the very high concentration of hemoglobin in blood.

Reviewer #2 (Remarks to the Author):

See attached PDF files with comments suggestions.

In their review of the first version of this manuscript, reviewer #2 added some comments to the manuscript file. These comments were forwarded to the authors, who replied as included in this Peer Review File.

Reviewer #2 Attachment on the following page

Non-invasive assessment of normal and impaired iron homeostasis in 2023 living human brains

REVIEWER'S COMMENTS

The work presented in the attached manuscript addresses an important issue of probing molecular environment of iron in the human brain in vivo through the use of conventional MRI measurements. By combining two common and clinically useful contrasts, the authors develop a new quantitative biomarker that is sensitive to iron homeostasis. The work has strong clinical motivation, and it includes several levels of validations such as the use of in vitro models, gene-expression, and in vivo on healthy brains and on volunteers with meningioma brain tumors.

Another important aspect of the work is a newfound approach for separating the effect of myelin and iron on MRI signals. These two factors affect MRI relaxation times conjointly, and a tool that can examine the state of iron in the tissue separately from myelin is highly important. Lastly, authors also assess reproducibility of the new biomarker across different brain regions.

Apart from the applicability shown in the current manuscript I believe that the new biomarker can be further utilized for other organs (e.g., liver) and other pathologies. Moreover, it joins an important list of quantitative MRI markers, and can thus be utilized to improve radiomic profiling of tissues in general.

I recommend publication, although some issues need to be addressed. These are delineated below, separated to Major and Minor comments.

MAJOR COMMENTS

- 1) Can the authors comment on the choice of using R1 and R2*, and the choice not using R2?

In the same context:

P11L174: authors state that r1-r2* relaxivity is less sensitive to myelin content. Glasser et al previously suggested an approach for mapping myelin using T1w/T2w imaging:

<https://doi.org/10.1523/JNEUROSCI.2180-11.2011>

<https://www.jneurosci.org/content/31/32/11597>

It is obvious that T1w/T2w is not the same as r1-r2* measures. Still, there is some underlying similarity between the two combinations. Can the authors comment on the differences between these two approaches and why is one less sensitive to myelin while the other is used as a proxy for mapping myelin content?

- 2) The study includes a large amount of human data which underwent several MRI scan-protocols. To improve readability, I would recommend adding a Table that delineates all healthy and patient cohorts and which MRI data sets / protocols were collected for each.

MINOR COMMENTS

- 3) Figure 1e: Colors are hard to distinguish – I would suggest adjusting the color palette to be more visually separable. The relaxivity model prediction for the Liposomal transferrin and Liposomal ferrous iron are outside the confidence interval for the experimental estimation of the r_1 - r_2^* relaxivity values. This should be discussed.
- 4) P8L117-119 – I would suggest reiterating (shortly) how / why is it possible to measure r_1 - r_2^* relaxivity in vivo as opposed to conventional iron relaxivity.
- 5) P26L506: Eq. → Eqs.
- 6) P26L509: please write the expressions for the constants explicitly as they do include $r_{(1,a)} / r_{(2,a)}$ which is not a simple constant.
- 7) P28L588: As far as I understand, b contains both $r_{(1,a)}$ and $r_{(2,a)}$. Please explain why can this term be considered a constant.
- 8) P26L517: “the in vivo iron relaxivity” → “the in vivo relaxivities” ?
- 9) P24: “ R_1 on R_2^* within an ROI in the”. Should this be “within a voxel” instead? Namely, doesn’t the suggested approach analyze relaxivity values per voxel?
- 10) P24: please state what are the units of T_f and F_t (units of concentration according to Eqs. S4 and S5).
- 11) Figure S19 caption describe four mixtures but the Figure shows six mixtures.
- 12) Figure S21 is too small to see – please provide a zoomed image.
- 13) P7L552: “For calibration” – please specific which calibration and also what acquisition scheme was used (apart from the preparation IR and post excitation spin-echo refocusing RF pulse).
- 14) P28L588 & L594: Equations are missing numbers.
- 15) P28L600-601 (and also relating to P22L405-407 and P22L414-416): MRI estimations of iron concentrations and relaxivities are done on a voxel level, producing an average value across each voxel. The iron concentrations calculated here, however, are based on content of ferritin and transferrin on a molecular length-scale. What is the interrelation between these two regimes? Can the molecular model apply to MRI-derived values which are averaged across relatively huge and heterogeneous MRI voxels?
- 16) P29L630: please provide a reason for collecting MT weighted data.
- 17) P31L675: please change B_1 to B_1^+ .
- 18) P32L708 and elsewhere throughout the manuscript: please use subscripts for R_1 and R_2^* where appropriate (i.e., R_1 and R_2^*).
- 19) P32L710-717 and P9L161-162: does this procedure of using a 5x5x5 moving window and relying on the local neighborhood of each voxel blur the ensuing r_1 - r_2^* maps?
- 20) P32L724: typo, remove the word “with”.

- 21) P9L145-147: why does the stability of r_1 - r_2^* relaxivity across different myelin content (in vitro model) indicate that r_1 - r_2^* relaxivity has high specificity to differences in molecular state of the iron?
- 22) Figure 2b: why does the left panel reflect the *reliability* of the method (as stated in the caption) rather than the variability of the measured values across different brain regions.
- 23) P11L164-173 and Figure S13: WM regions are clearly separated using the newly suggested relaxivity biomarker. This is shown on a group-wise level in Figure S13. Does this also hold on a single subject level?
- 24) P12L206: R^2 of 0.59 is reported while Figure 3c shows a value of 0.62.
- 25) Figure 4: shows R_1 and R_2^* maps. Please also add a map of the r_1 - r_2^* relaxivity parameter of interest.
- 26) Figure 4c-f: please mark the tumor ROI which was included in the analysis that is shown in these panels.
- 27) P18L278-279: $p > 0.01$ is considered as not statistically correlated. Conventionally, 0.05 is set as a statistically significant limit. Please add the specific p-value and elaborate why it is not considered significant.
- 28) P19L314: commonly is → is commonly
- 29) P21L362: farther → further
- 30) Text that can use rephrasing to improve readability is marked with green underline in the attached PDF file.

Dear reviewers,

Thank you for agreeing to review the manuscript entitled " Non-invasive assessment of normal and impaired iron homeostasis in living human brains". We highly appreciated your thoughtful comments, which we believe improved the manuscript substantially.

We revised the manuscript according to the reviewers' suggestions. The revised version includes new in vitro experiments, new in vivo analyses, and new numerical simulations. Taken together, these new analyses all provide further validation for the sensitivity of the r_1 - r_2^* relaxivity approach to brain iron homeostasis. The manuscript has gone through considerable adjustments, including changes to the text as well as several additional supplementary sections. Below, we describe the changes made to the manuscript following each comment. The comments are in blue text, our responses are in black, and the revised text is in gray. Changes are highlighted in yellow in the revised manuscript.

Reviewer #1 (Remarks to the Author):

In this new manuscript, Filo et al use a relation between R_1 and R_2^* relaxivity, which are MRI parameters sensitive to iron, to estimate tissue concentrations in brain of transferrin and ferritin – the major non-heme binding proteins. In so doing, they present a new contrast that may have utility in different settings. The approach they investigate is brain cancer, where they show the advantage over other techniques. The authors begin by characterising the signals for transferrin and ferritin in vitro, before applying this to human brains, and referencing these changes to transcriptional profiles and historical measures of brain iron. These are novel and important findings that may yet be further developed to investigate other iron binding partners (heme?) and applied in different contexts. While there will inevitably be questions (that may not be answerable) as to the validity of this measure for certain species of iron, the authors have developed their argument well, and they have introduced a new contrast for the field regardless.

We thank the reviewer for appreciating our work and finding it novel and important.

I have the following questions:

1. both R_1 and R_2^* have the same unit. why not perform a voxel-wise division? The current approach where each ROI is binned followed by a regression fitting seems to smooth the data and gather all the important contributions of other sources in the constant.

We thank the reviewer for this interesting observation regarding the R_1/R_2^* ratio, which highlights an important aspect of our work (see discussion on the important contributions of the constant in the next point). We agree with the reviewer that the voxel-wise division of R_1 and R_2^* (the R_1/R_2^* ratio) has the benefit of allowing higher spatial resolution. However, we demonstrate that the R_1/R_2^* ratio has a different biophysical interpretation compared to the r_1 - r_2^* relaxivity. Mainly, the R_1/R_2^* ratio depends on the relative magnitudes of R_1 and R_2^* . On the other hand, the slope of the R_1 - R_2^* linear fit (i.e. the r_1 - r_2^* relaxivity) represents the change in R_1 relative to the change in R_2^* ($\Delta R_1/\Delta R_2^*$), and is therefore less sensitive to the magnitude of these relaxation rates and more sensitive to their shared variation. To

demonstrate this point, we performed simulations of the R_1/R_2^* ratio and the $r_1-r_2^*$ relaxivity across the physiological ranges of transferrin-ferritin fraction, iron compounds

Sup. Figure 39: Simulations of the sensitivity of the R_1/R_2^* ratio and the $r_1-r_2^*$ relaxivity to different biophysical sources. The variability in R_1/R_2^* and $r_1-r_2^*$ relaxivity was tested across the physiological ranges of transferrin-ferritin fraction (a), iron compounds concentration (b), myelin concentration (c) and myelin variability (ΔM , d). In each panel only one biological property is changing while the rest are kept fixed. Simulations were performed under the same framework as in Supplementary Section 4.3.

concentration, myelin concentration and myelin variability (ΔM). In each simulation we changed one biological component while keeping the rest fixed. These simulations follow the same framework presented in Supplementary Section 4.3. Sup. Figure 39 demonstrates that the R_1/R_2^* ratio changes with several physiological properties; the transferrin-ferritin fraction, the iron compounds concentration and the myelin concentration. On the other hand, the $r_1-r_2^*$ relaxivity changes mostly with the transferrin-ferritin fraction. It is also sensitive, but to a smaller degree, to changes in ΔM (Supplementary Section 4.3). Changes in the myelin and the iron compounds concentrations do not affect the simulated $r_1-r_2^*$ relaxivity. Therefore, the R_1/R_2^* ratio and the $r_1-r_2^*$ relaxivity have different biophysical sources.

To further demonstrate the differences between the $r_1-r_2^*$ relaxivity and the R_1/R_2^* ratio, we provide a comparison between these measurements *in vivo*. **Sup. Figure 40** shows that across subjects and brain regions, the $r_1-r_2^*$ relaxivity is different compared to the R_1/R_2^* ratio. **Sup. Figure 41** emphasizes that the contrast of the R_1/R_2^* ratio across the brain is different from the $r_1-r_2^*$ relaxivity contrast (shown in **Figure 2b**). Namely, the contrast between different white-matter regions that was observed in the $r_1-r_2^*$ relaxivity (**Sup. Figure 13**) vanishes in the R_1/R_2^* ratio. The contrast between different gray-matter regions is also different between these measurements. Another example for the differences between the measurements is

Sup. Figure 40: The $R1/R2^*$ ratio across the brain. The variation across young subjects (age 27 ± 2 , $N = 21$) in the $R1/R2^*$ ratio in different brain regions. The 25th, 50th and 75th percentiles and extreme data points are shown for each box.

Sup. Figure 41: comparison between the $R1/R2^*$ ratio (y-axis) and the $r1-r2^*$ relaxivity (x-axis) across different brain areas (different colors) for all healthy human subjects ($N=39$).

that the hippocampus, which has the second lowest $r1-r2^*$ relaxivity, is similar to the thalamus and is closer to white-matter regions in terms of its $R1/R2^*$ ratio.

These analyses validate that the $r1-r2^*$ relaxivity is inherently distinct from the $R1/R2^*$ ratio. It highlights the unique nature of the relaxivity measurement, which does not depend on the magnitude of the relaxation rates but rather on their shared variation. This reduces the effect of the myelin and iron concentrations on the $r1-r2^*$ relaxivity, and enhances its sensitivity to the iron homeostasis.

This discussion is now included in the supplemental material (Supplementary Section 10).

It is also mentioned in the main text (lines 177-180): “Importantly, the $R1/R2^*$ ratio may be more similar to the $r1-r2^*$ relaxivity than the individual measurements of $R1$ and $R2^*$. Yet, in vivo estimation of the $R1/R2^*$ ratio show it is uncorrelated with the $r1-r2^*$ relaxivity and does not separate different white-matter regions (Supplementary Section 10).

And lines 194-199: As in the in vivo brain, we found that changes in the myelin concentration substantially affect the simulated measurements of $R1, R2^*$ (Supplementary Section 4.3) and the $R1/R2^*$ ratio (Supplementary Section 10). However, myelin-related changes were not the main component governing the simulated measurement of the $r1-r2^*$ relaxivity, and in simulations of physiological conditions they could not explain the variability in the $r1-r2^*$ relaxivity across the brain (Supplementary Section 4.3)“

And in the discussion (lines 379-384): “The $R1/R2^*$ ratio may be more similar to the $r1-r2^*$ relaxivity than the individual measurements of $R1$ and $R2^*$. However, the $R1/R2^*$ ratio is sensitive to the relative magnitudes of $R1$ and $R2^*$, and the $r1-r2^*$ relaxivity is only affected by the shared variation of $R1$ and $R2^*$. We demonstrate in vivo that these measurements are different. In simulations we find that unlike the $r1-r2^*$ relaxivity, the $R1/R2^*$ ratio is sensitive to iron compounds concentration and myelin content.”

2. It would also be interesting to look at the intercept of the fit.

We agree with the reviewer that the intercept of the fit has an interesting biophysical interpretation and believe that a discussion on this point will contribute to the manuscript. The intercept of the R_1 - R_2^* linear fit represents the residual R_1 not explained by R_2^* . Therefore, this measurement has the potential to be sensitive to biological sources affecting exclusively R_1 and not R_2^* . Based on the relaxivity model (“*In vivo* iron relaxivity model” in methods), the expression for intercept can be deduced:

$$R_1 = r_{(1,a)}[a] + c_{(1,a)}$$

$$R_2^* = r_{(2,a)}[a] + c_{(2,a)}$$

Substituting Eq. 2 in Eq. 1:

$$R_1 = \frac{r_{(1,a)}}{r_{(2,a)}} R_2^* + c_a$$

Where the intercept can be expressed as:

$$c_{a/b} = c_{(1,a/b)} - \frac{r_{(1,a/b)}}{r_{(2,a/b)}} c_{(2,a/b)}$$

Therefore, it should be sensitive both to the non-iron contributions to R_1 and R_2^* ($c_{(1/2,a/b)}$), and to their iron relaxivities ($\frac{r_{(1,a/b)}}{r_{(2,a/b)}}$). We first tested the biophysical interpretation of the intercept *in vitro*. **Sup.**

Sup. Figure 32: The intercept of the R_1 - R_2^* linear fit *in-vitro*. The intercept of the R_1 - R_2^* linear fit for different iron environments. For each box, the central mark is the intercept, and the box shows the 95% confidence bounds of the linear fit.

liposomal transferrin is more significant when measuring the r_1 - r_2^* relaxivity ($p < 10^{-8}$, Bonferroni corrected ANCOVA test).

Figure 32 shows the intercept of different iron forms. Interestingly, the intercept seems to be less sensitive to the molecular environment of the iron forms compared to the r_1 - r_2^* relaxivity. For example, the r_1 - r_2^* relaxivity varied greatly between free and liposomal ferritin (Figure 1e, $p < 10^{-5}$, Bonferroni corrected ANCOVA test). On the other hand, the intercept is not significantly different between free and liposomal ferritin ($p = 0.99$, Bonferroni corrected ANCOVA test). The intercept is sensitive to the molecular form of iron, and is different for liposomal ferritin and liposomal transferrin ($p = 0.0036$, Bonferroni corrected ANCOVA test). Nevertheless, the difference between liposomal ferritin and

Next, we tested the intercept of the R_1 - R_2^* linear fit *in vivo*. **Sup. Figure 33a** shows the R_1 - R_2^* linear fit for four brain regions of a single subject. **Sup. Figure 33b** shows the variability of the intercept across young subjects for different brain regions. Interestingly, while white-matter regions tend to have high R_1 , R_2^* and r_1 - r_2^* relaxivity values, they have low intercept. Therefore, in the white-matter, most of the variability in R_1 is explained by R_2^* , implying for shared biological sources that govern both relaxation mechanisms in this tissue. On the contrary, gray-matter and subcortical regions have higher intercept, suggesting that in these regions there is a residual R_1 relaxation not explained by R_2^* . This residual R_1 could be attributed to biological sources affecting R_1 exclusively.

R_1 relaxation mechanisms are affected by local molecular interactions, while R_2^* is sensitive to more global effects of extended paramagnetic interactions at the mesoscopic scale³⁷. Therefore, it could be that the intercept is more sensitive to local molecular interactions that do not involve extended paramagnetic effects. Further work may provide insights into biological substrates that are characterized by such relaxation. To conclude, while we exploit the slope of the R_1 - R_2^* fit (r_1 - r_2^* relaxivity) for information on the iron homeostasis, the intercept might capture important information as well. Analyzing both the relaxivity and the intercept may contribute complementary information and allow better *in vivo* characterization of brain tissue.

Sup. Figure 33: The intercept of the R_1 - R_2^* linear fit *in-vivo*. (a) The dependency of R_1 on R_2^* in four representative brain regions (WM-occipital, CTX-occipital, Thalamus & Putamen) of a single subject. R_2^* and R_1 were binned (dots represent the median; shaded areas represent the mean absolute deviation), and a linear fit was calculated. The slopes and intercepts of the linear fit vary across brain regions. (b) The intercept of the R_1 - R_2^* fit across the brain. Left: the reliability of the method in different brain regions as observed by the variation in the intercept across normal subjects (age 27 ± 2 , $N = 21$). The 25th, 50th and 75th percentiles and extreme data points are shown for each box. Right: the contrast of the intercept across the brain. Red, yellow and gray distributions represent the values of the intercept in sub-cortical (sub-CTX), white-matter (WM) and cortical (CTX) brain regions, respectively.

This discussion is now included in the supplemental material (Supplementary Section 6).

It is mentioned in the main text (lines 162-163): “While the r_1 - r_2^* relaxivity represents the slope of the R_1 - R_2^* linear fit, we also find that the intercept varies across the brain (Supplementary Section 6).”

And in the discussion (lines 439-442): “*The intercept of the R_1 - R_2^* linear fit represents the residual R_1 not explained by R_2^* . We find that the relaxivity and the intercept contribute complementary information both in vivo and in vitro. Therefore, combining these measurements may allow better in vivo characterization of brain tissue*”

The exact expression for the intercept is now presented in the methods section “In vivo iron relaxivity model” (lines 527-529).

The r_1 - r_2^* relaxivity approach assumes the R_1 and R_2^* signal is only coming from non-heme (ferritin and Transferrin) paramagnetic sources (or the extended version from Myelin). In in vivo tissues, there is a large contribution from iron from hemoglobin. The current resolution of imaging may not allow to capture this. Could an in vitro experiment demonstrate that r_1 - r_2^* relaxivity of ferritin or transferrin is not impacted by hemoglobin when it is added? – possibly at 10 fold greater concentrations to reflect the very high concentration of hemoglobin in blood.

We thank the reviewer for raising this point, and believe that discussing the effect of hemoglobin on the r_1 - r_2^* relaxivity will contribute to the biophysical understanding of our new approach. To address the reviewer’s comment, we performed a new in-vitro experiment. First, we assessed the r_1 - r_2^* relaxivity of hemoglobin. Next, we examined whether the r_1 - r_2^* relaxivities of ferritin and transferrin change in the presence of hemoglobin (two different hemoglobin concentrations were tested). The hemoglobin concentrations in this experiment were constrained by its solubility (20mg/mL for hemoglobin alone, and lower when it is in a mixture with other iron-binding proteins). **Sup. Figure 34a-b** shows the dependency of R_1 and R_2^* on the iron compound concentration. Hemoglobin has a distinct R_1 relaxivity ($p < 0.0001$ when comparing hemoglobin to all other samples, bonferroni-corrected ANCOVA test), and an R_2^* relaxivity similar to that of transferrin ($p < 0.0001$ when comparing hemoglobin to ferritin and ferritin-hemoglobin mixtures, $p = n.s$ for comparing to transferrin and transferrin-hemoglobin mixtures, bonferroni-corrected ANCOVA test). Importantly, while adding hemoglobin to ferritin and transferrin affects R_1 and R_2^* values, the relaxivities of both ferritin and transferrin are not statistically affected by the presence of hemoglobin (**Sup. Figure 34a-b**, $p > 0.05$ for comparing transferrin to transferrin-hemoglobin mixtures and ferritin to ferritin-hemoglobin mixtures, bonferroni-corrected ANCOVA test). Evaluating the r_1 - r_2^* relaxivities, we find that hemoglobin has lower values similar to ferritin (**Sup. Figure 34c**). Importantly, we compared this result with values previously published in the literature. Blockley et al. evaluated the dependency of whole blood R_1 and R_2^* values on the deoxyhemoglobin concentrations⁴⁰. We divided the R_1 and R_2^* relaxivity of deoxyhemoglobin to get an estimation for the r_1 - r_2^* relaxivity (eq. 5). The prediction from the literature is presented in a red data point in **Sup. Figure 34d**. We find great agreement between our estimation of the hemoglobin r_1 - r_2^* relaxivity, and the literature reported values. Next, we tested whether the r_1 - r_2^* relaxivities of ferritin and transferrin are affected by the presence of hemoglobin (**Sup. Figure 34**). We found that even in hemoglobin mixtures, ferritin and transferrin have distinct relaxivities ($p < 0.005$ when comparing all ferritin-containing samples to all

Sup. Figure 34: the effect of hemoglobin on the r_1 - r_2^* relaxivity. (a-b) The dependency of R_1 and R_2^* on the iron compound concentrations for different iron environments: free ferritin, free transferrin, free hemoglobin, ferritin-hemoglobin and transferrin-hemoglobin mixtures (in two different hemoglobin concentration: 5 & 10 mg/ml). For mixtures- data points represent samples with varying ferritin/transferrin concentrations while hemoglobin concentrations were kept constant (x-axis shows total iron compound concentrations including hemoglobin). The linear relationships between relaxation rates and iron compounds concentrations are marked by lines. We define the slopes of these lines as the iron relaxivities. Dashed lines represent extrapolation of the linear fit. Shaded areas represent the 95% confidence bounds. **(c)** The iron relaxivities of R_1 and R_2^* for ferritin and transferrin in the presence of hemoglobin. Iron relaxivities are calculated by taking the slopes of the linear relationships shown in (a,b), and are measured in $[\text{sec}^{-1}/(\text{mg}/\text{ml})]$. For each box, the central mark is the iron relaxivity (slope); the box shows the 95% confidence bounds of the linear fit. **(d)** The dependency of R_1 on R_2^* for ferritin and transferrin in the presence of hemoglobin. Data points represent samples with varying iron compound concentrations. The linear relationships of R_1 and R_2^* are marked by lines. The slopes of these lines are the r_1 - r_2^* relaxivities. Dashed lines represent extrapolation of the linear fit. Shaded areas represent the 95% confidence bounds. **(e)** The r_1 - r_2^* relaxivities in the presence of hemoglobin. For each box, the central mark is the r_1 - r_2^* relaxivity, and the box shows the 95% confidence bounds of the linear fit. Red dot indicates the deoxyhemoglobin r_1 - r_2^* relaxivity estimated from the ratio between the R_1 and R_2^* relaxivities of deoxyhemoglobin in blood reported by Blockley et al.⁴⁰. Even in the presence of hemoglobin, ferritin and transferrin have distinct r_1 - r_2^* relaxivities ($p < 0.005$ when comparing all ferritin-containing samples to all transferrin-containing samples, bonferroni-corrected ANCOVA test).

transferrin-containing samples, bonferroni-corrected ANCOVA test). Adding hemoglobin to ferritin and transferrin did not have significant effect on the r_1 - r_2^* relaxivity ($p > 0.05$ bonferroni-corrected ANCOVA test). Notably, we scanned the hemoglobin samples twice, immediately after the preparation and a week

later. While the rest of our *in vitro* experiments were stable and reproducible for different scan times, hemoglobin samples scanned a week after the preparation showed aggregation, visible both by eye and in the MRI scans. This aggregation was less visible in the first scan, immediately after the preparation. Therefore, we used results from the first scan for the analysis. However, aggregation processes may still impact these results.

Importantly, these *in vitro* experiments may not capture fully the effect of hemoglobin on the $r_1-r_2^*$ relaxivity in the *in vivo* brain. Particularly, the reviewer rightly points out that the hemoglobin concentration in the blood is extremely high. We can simulate to what extent hemoglobin can affect the *in vivo* $r_1-r_2^*$ relaxivity. In normal gray-matter and white-matter, approximately 4–6% and 1–3% of the tissue volume is occupied by blood⁹³. Assuming an extreme scenario where all blood volume is occupied by deoxyhemoglobin, and that the $r_1-r_2^*$ relaxivity of deoxyhemoglobin in blood is as reported in the literature⁴⁰, the $r_1-r_2^*$ relaxivity effect in a voxel with 6% blood volume would be:

$$\text{hemoglobin } r_1 - r_2^* \text{ relaxivity} * \text{blood volume} = 7.2 * 10^{-4} * 0.06 = 4.3 * 10^{-5}$$

This is less than 1% percent of the average $r_1-r_2^*$ relaxivity measured in the *in vivo* brain. Therefore, based on this simulation, hemoglobin is not expected to be the main source governing the $r_1-r_2^*$ relaxivity contrast in the brain. Nonetheless, it could be that some of the $r_1-r_2^*$ relaxivity effects that we measured in the brain are related to hemoglobin. We therefore believe that the $r_1-r_2^*$ relaxivity in the brain is probably not sensitive exclusively to ferritin and transferrin. Hemoglobin and many other aspects of the iron homeostasis could be reflected in the $r_1-r_2^*$ relaxivity measurement.

This new experiment is now described in the supplemental material (Supplementary Section 7).

It is mentioned in the main text (lines 143-145): “*Hemoglobin affects the R1 and R2* relaxivities of blood*⁴⁰. *In vitro*, we found that ferritin and transferrin have distinct $r_1-r_2^*$ relaxivities even in the presence of hemoglobin (Supplementary Section 7).”

And in the discussion (lines 411-436): “*Our in vitro experiments are based on ferritin, transferrin and ferrous iron as examples for variable iron environments. Yet, in the human brain our approach is probably more broadly sensitive to the entire milieu of iron compounds, their iron binding characteristics and aggregation states. Other iron compounds that exist in the brain, such as hemoglobin, hemosiderin, neuromelanin, magnetite, ferric ion, lactoferrin and melanotransferrin*⁴, *might have distinct iron relaxivities as well (see Supplementary Section 7 for the iron relaxivity of hemoglobin). ... In addition, the paramagnetic properties of deoxyhemoglobin in capillaries and veins and their orientations are known to affect the R2* measurement*⁶⁷⁻⁶⁹. *In vitro*, we found that ferritin and transferrin have distinct $r_1-r_2^*$ relaxivities even in the presence of hemoglobin (Supplementary Section 7). However, hemoglobin could have a more substantial effect in vivo. Thus, the $r_1-r_2^*$ relaxivity might also be sensitive to the iron environment in the vascular system, which is crucial for brain iron metabolism and homeostasis⁷⁰. ... To conclude, while we demonstrate our approach in vitro based on ferritin, transferrin

and ferrous iron samples, we believe that in the human brain the r_1 - r_2^ relaxivity provides more comprehensive information on the molecular iron milieu previously inaccessible by MRI. “*

Reviewer #2

The work presented in the attached manuscript addresses an important issue of probing molecular environment of iron in the human brain in vivo through the use of conventional MRI measurements. By combining two common and clinically useful contrasts, the authors develop a new quantitative biomarker that is sensitive to iron homeostasis. The work has strong clinical motivation, and it includes several levels of validations such as the use of in vitro models, gene-expression, and in vivo on healthy brains and on volunteers with meningioma brain tumors. Another important aspect of the work is a newfound approach for separating the effect of myelin and iron on MRI signals. These two factors affect MRI relaxation times conjointly, and a tool that can examine the state of iron in the tissue separately from myelin is highly important. Lastly, authors also assess reproducibility of the new biomarker across different brain regions. Apart from the applicability shown in the current manuscript I believe that the new biomarker can be further utilized for other organs (e.g., liver) and other pathologies. Moreover, it joins an important list of quantitative MRI markers, and can thus be utilized to improve radiomic profiling of tissues in general. I recommend publication, although some issues need to be addressed. These are delineated below, separated to Major and Minor comments.

We thank the reviewer for finding our work important and valuing its wide applicability to the field of MRI. Following the insightful comments of the reviewer, we revised the manuscript and performed several new analyses, as described below.

MAJOR COMMENTS

- 1) Can the authors comment on the choice of using R_1 and R_2^* , and the choice not using R_2 ?

The reviewer raises an interesting point. Following the reviewer's comment we acknowledged that not only R_2^* but also R_2 is sensitive to the effects of iron on the transverse relaxation^{3,4}. Theoretically, the dependency of R_1 on R_2 could also be used as an in-vivo estimation of the iron relaxivity. In order to follow the reviewer's suggestion, we added to the revised manuscript R_2 measurements of different iron environments. This new analysis (**Sup. Figure 35**) shows that different iron environments have distinct R_2 -iron relaxivity ($p(\text{ANCOVA}) < 10^{-70}$). Comparing the R_2 -iron relaxivity (**Sup. Figure 35**) and the R_2^* -iron relaxivity (**Figure 1c**) we find similar contrast between iron environments (e.g. highest relaxivity for ferrous iron and lowest relaxivity for transferrin). This contrast is different from the R_1 -iron relaxivity. For example, while the R_1 -iron relaxivity of ferritin is similar for free and liposomal states (**Figure 1c**), both the R_2 -iron relaxivity and the R_2^* -iron relaxivity are different for liposomal and free ferritin (**Sup. Figure 35a-b, Figure 1c**). Therefore, these results suggest that R_2 could be used instead of R_2^* when calculating the relaxivity. To test this, we estimated the r_1 - r_2 relaxivity for different iron environments (**Sup. Figure 35c-d**). We found that different iron environments have distinct r_1 - r_2 relaxivity ($p(\text{ANCOVA}) < 10^{-39}$). Moreover, we found great agreement between the experimental results and the theoretical predictions for the r_1 - r_2 relaxivity based on the ratio of iron relaxivities (eq. 5). Interestingly, while the theoretical prediction for the r_1 - r_2^* relaxivity of ferrous iron and liposomal transferrin was a bit higher than the

Sup. Figure 35: The dependency of R1 on R2 for assessing different iron environments. (a) The dependency of R2 on the iron compound concentrations for different iron environments: free ferritin, liposomal ferritin, bovine serum albumin (BSA)-ferritin mixture, free transferrin, liposomal transferrin and liposomal ferrous iron. Data points represent samples with varying concentrations relative to the water fraction ([mg/wet ml]). The linear relationships between relaxation rates and iron- compounds concentrations are marked by lines. We define the slopes of these lines as the iron relaxivities. Dashed lines represent extrapolation of the linear fit. Shaded areas represent the 95% confidence bounds. **(b)** The iron relaxivity of R2 is different for different iron environments ($p(\text{ANCOVA}) < 10^{-70}$). Iron relaxivities are calculated by taking the slopes of the linear relationships shown in (a,b), and are measured in $[\text{sec}^{-1}/(\text{mg/wet ml})]$. For each box, the central mark is the iron relaxivity (slope); the box shows the 95% confidence bounds of the linear fit. **(c)** The dependency of R1 on R2 for different iron environments. Data points represent samples with varying iron compound concentrations relative to the water fraction. The linear relationships of R1 and R2 are marked by lines. The slopes of these lines are the r1-r2 relaxivities. Dashed lines represent extrapolation of the linear fit. Shaded areas represent the 95% confidence bounds. **(d)** The r1-r2 relaxivities are different for different iron environments ($p(\text{ANCOVA}) < 10^{-39}$). For each box, the central mark is the r1-r2 relaxivity, and the box shows the 95% confidence bounds of the linear fit. Red dots indicate the successful prediction of the experimental r1-r2 relaxivity from the ratio between the iron relaxivities of R1 and

experimental results (**Figure 1e**), the theoretical prediction of these experimental results is more accurate for the r_1 - r_2 relaxivity. Therefore, we believe that estimating the r_1 - r_2 relaxivity may also be a useful marker of the iron homeostasis. Technically, acquiring R_1 and R_2^* with standard MRI protocols can be shorter than acquiring R_1 and R_2 . Both R_1 and R_2^* can be estimated from the same widely available sequence (spoiled gradient echo with multiple echoes). However, if measurements of both R_1 and R_2 are available, we believe that an estimation of the r_1 - r_2 relaxivity also could contribute to the *in vivo* characterization of the iron homeostasis. By joining several iron relaxivity measurements, it might be possible to further disentangle the contributions of different iron environments to the MRI signal.

These analyses are now discussed in the supplemental material (Supplementary Section 8).

They are mentioned in the discussion (lines 437-451): “*In order to further model the separate contributions to the MRI signal of different iron compounds, it would be necessary to increase the dimensionality of the in vivo iron relaxivity measurement. In addition to R_1 and R_2^* , other qMRI parameters known to be sensitive to iron include quantitative susceptibility mapping (QSM) and $R_2^{3,4}$. In addition, it was suggested that magnetization transfer (MT) measurements are affected by neuromelanin-iron complexes^{12,73}. The linear interdependencies of these other iron-related MRI measurements may uncover additional features of the iron environment⁷⁴. For example, we show that the dependency of R_1 on R_2 is also useful for differentiating between iron environments. Therefore, we speculate that the concept we introduce here, of exposing the iron relaxivities in vivo based on the linear dependency of R_1 on R_2^* (the r_1 - r_2^* relaxivity), can be generalized to further increase MRI’s specificity for iron using additional complementary measurements.*”

And in the results (lines 125-128): “*For liposomal transferrin and ferrous iron, the model’s prediction is slightly outside the confidence interval for the experimental results. This seems to be related to R_2^* estimations, as the prediction improves when replacing R_2^* with R_2 (Supplementary Section 8).*”

In the same context: P11L174: authors state that r_1 - r_2^* relaxivity is less sensitive to myelin content. Glasser et al previously suggested an approach for mapping myelin using T1w/T2w imaging: <https://doi.org/10.1523/JNEUROSCI.2180-11.2011> <https://www.jneurosci.org/content/31/32/11597>

It is obvious that T1w/T2w is not the same as r_1 - r_2^* measures. Still, there is some underlying similarity between the two combinations. Can the authors comment on the differences between these two approaches and why is one less sensitive to myelin while the other is used as a proxy for mapping myelin content?

We thank the reviewer for raising this concern. As the semi-quantitative T1w/T2w imaging is very popular, we believe that a clarification on the differences between this myelin marker and the r_1 - r_2^* relaxivity is important and should be incorporated in the manuscript. Both the r_1 - r_2^* relaxivity and the T1w/T2w approaches represent combinations of transverse and longitudinal relaxation. Mathematically, it can be shown that the T1w/T2w enhances the myelin contribution, while the r_1 - r_2^* relaxivity reduces the myelin contribution. T1w is proportional to R_1 , and T2w is proportional to $1/R_2$. Therefore, T1w/T2w is

Sup. Figure 36: Simulations of the sensitivity of $R_1 \cdot R_2^*$ (a-d) and $r_1 - r_2^*$ relaxivity (e-h) to different biophysical sources. The variability in $R_1 \cdot R_2^*$ and $r_1 - r_2^*$ relaxivity was tested across the physiological ranges of transferrin-ferritin fraction (a,e), myelin concentration (b,f), myelin range (ΔM , c,g) and iron compounds concentrations (d,h). Simulations were performed under the same framework as in Supplementary Section 4.3. In each simulation we changed one biological component while keeping the rest fixed.

proportional to $R_1 \cdot R_2$. Assuming both R_1 and R_2 are linearly related to myelin, then $T1w/T2w$ is proportional to myelin² (as argued by Glasser et al⁵⁵). On the contrary, the $r_1 - r_2^*$ relaxivity (i.e., the slope of the $R_1 - R_2^*$ linear fit) represents the change in R_1 relative to the change in R_2^* ($\Delta R_1 / \Delta R_2^*$), and is therefore less sensitive to the magnitude of these relaxation rates and more sensitive to their shared variation. While the myelin concentration has a large effect on the magnitudes of R_1 and R_2^* , its effect of the shared variation of R_1 and R_2^* ($\Delta R_1 / \Delta R_2^*$) is minimal. $\Delta R_1 / \Delta R_2^*$ is mostly related to the variability in myelin concentration within ROIs (ΔM) (Supplementary Section 4.3). We show in simulations that non-physiological values of ΔM are required to produce the $r_1 - r_2^*$ relaxivity values measured in the brain. To demonstrate the differences between $T1w/T2w$ and the $r_1 - r_2^*$ relaxivity, we performed simulations of $r_1 - r_2^*$ relaxivity and $R_1 \cdot R_2^*$ (similar to $R_1 \cdot R_2$ which is proportional to $T1w/T2w$). We tested these parameters across the physiological ranges of transferrin-ferritin fraction, myelin concentration, myelin range (ΔM) and iron compounds concentrations. In each simulation we changed one biological component while keeping the rest fixed. These simulations follow the same framework presented in Supplementary Section 4.3. **Sup. Figure 36** demonstrates that $R_1 \cdot R_2^*$ changes mostly with the myelin and iron concentrations, but also with the transferrin-ferritin fraction. On the other hand, the $r_1 - r_2^*$ relaxivity changes mostly with

the transferrin-ferritin fraction. It also changes with the myelin variability (ΔM), but to a smaller extent (Supplementary Section 4.3). Nevertheless, the simulated $r_1-r_2^*$ relaxivity does not change with the myelin and iron concentrations.

Moreover, we provide a comparison between the $r_1-r_2^*$ relaxivity and $R_1 \cdot R_2^*$ *in vivo*. **Sup. Figure 37** shows that across subjects and brain regions, the $r_1-r_2^*$ relaxivity is different compared to $R_1 \cdot R_2^*$. **Sup. Figure 38** emphasizes that the contrast of $R_1 \cdot R_2^*$

Sup. Figure 37: comparison between $R_1 \cdot R_2^*$ (y-axis) and the $r_1-r_2^*$ relaxivity (x-axis) across different brain areas (different colors) for all healthy human subjects ($N=39$).

Sup. Figure 38: $R_1 \cdot R_2^*$ across the brain. The variation across young subjects (age 27 ± 2 , $N = 21$) in $R_1 \cdot R_2^*$ in different brain regions. The 25th, 50th and 75th percentiles and extreme data points are shown for each box.

across the brain is different from the $r_1-r_2^*$ relaxivity contrast (shown in **Figure 2b**). Interestingly, the contrast between white-matter regions shown in the $r_1-r_2^*$ relaxivity (**Figure 2b**), is not present in $R_1 \cdot R_2^*$. To conclude, T1w/T2w images are similar in contrast to $R_1 \cdot R_2^*$. We show mathematically, in simulations and *in vivo*, that $R_1 \cdot R_2^*$ is different from the $r_1-r_2^*$ relaxivity, and more sensitive to myelin concentration.

We now discuss the comparison of the $r_1-r_2^*$ relaxivity and T1w/T2w imaging in the manuscript (Supplementary Section 9). This issue is also mentioned in the main text:

Lines 186-188: “We also find that T1w/T2w contrast, which serves as a semi-quantitative myelin marker⁵⁵, is different from the $r_1-r_2^*$ relaxivity (Supplementary Section 9).”

Lines 372-373: “The semi-quantitative T1w/T2w myelin marker⁵⁵ is also not correlated with the $r_1-r_2^*$ relaxivity”

2) The study includes a large amount of human data which underwent several MRI scan-protocols. To improve readability, I would recommend adding a Table that delineates all healthy and patient cohorts and which MRI data sets / protocols were collected for each.

A table summarizing the human MRI datasets was added to the manuscript (see methods section under “Human MRI datasets”).

MINOR COMMENTS

3) Figure 1e: Colors are hard to distinguish – I would suggest adjusting the color palette to be more visually separable.

We thank the reviewer for this comment which improved the readability of the manuscript. We changed the color palette for the main figure and the supplemental figures showing in vitro results.

The relaxivity model prediction for the Liposomal transferrin and Liposomal ferrous iron are outside the confidence interval for the experimental estimation of the $r1-r2^*$ relaxivity values. This should be discussed.

We agree with the reviewer that this is an important point. We now discuss the prediction for liposomal transferrin and ferrous iron in the manuscript (lines 125-128): “*For liposomal transferrin and ferrous iron, the model’s prediction is slightly outside the confidence interval of the experimental results. This seems to be related to $R2^*$ estimations, as the prediction improves when replacing $R2^*$ with $R2$ (Supplementary Section 8).*”

4) P8L117-119 – I would suggest reiterating (shortly) how / why is it possible to measure $r1-r2^*$ relaxivity in vivo as opposed to conventional iron relaxivity.

This section was now revised based on the reviewer’s suggestion (lines 117-120): “*The iron relaxivity represents the dependency of relaxation on the iron concentration, which cannot be estimated in vivo. The $r1-r2^*$ relaxivity is defined as the dependency of $R1$ on $R2^*$, and thus only relies on MRI measurements that can be estimated in vivo.*”

5) P26L506: Eq. à Eqs. Done.

6) P26L509: please write the expressions for the constants explicitly as they do include $r(1,a) / r(2,a)$ which is not a simple constant.

We now write explicitly the expression for the constants (lines 528-529):

“*The intercept of the linear dependency of $R1$ on $R2^*$ can be described by the following expression:*

$$c_{a/b} = c_{(1,a/b)} - \frac{r_{(1,a/b)}}{r_{(2,a/b)}} c_{(2,a/b)}.$$

7) P28L588: As far as I understand, b contains both $r(1,a)$ and $r(2,a)$. Please explain why can this term be considered a constant.

We agree with the reviewer that the definition of b as a constant was not clear and should be clarified. b is the intercept of the $R1-R2^*$ linear fit. It therefore represents the residual $R1$ not explained by $R2^*$. The full expression for b based on our biophysical model includes $r(1,a)$ and $r(2,a)$. Yet, it does not change with $R2^*$ or $R1$ values and is therefore constant across all samples used for the linear fit. We revised this section (lines 618-619): “ *b is the intercept (the residual $R1$ not explained by $R2^*$).*”

Following the suggestion of reviewer 1, we also include a new discussion on the biophysical meaning of the $R1-R2^*$ intercept (Supplementary Section 6). This is also mentioned in the discussion (lines 439-442): “*The intercept of the $R1-R2^*$ linear fit represents the residual $R1$ not explained by $R2^*$. We find that the relaxivity and the intercept contribute complementary information both in vivo and in vitro. Therefore, combining these measurements may allow better in vivo characterization of brain tissue*”

8) P26L517: “the in vivo iron relaxivity” à “the in vivo relaxivities” ? Done.

9) P24: “R1 on R2* within an ROI in the”. Should this be “within a voxel” instead? Namely, doesn’t the suggested approach analyze relaxivity values per voxel?

This comment refers to supplementary section 4 which presents a biophysical model for the r1-r2* relaxivity.

First, we would like to clarify more generally, that the r1-r2* relaxivity in the brain is estimated on the ROI level. The r1-r2* relaxivity is based on the linear slope of R1 vs. R2*. As there is only one estimation of R1 and R2* per voxel, several voxels are needed in order to perform the linear fit. This is mentioned in the results section (lines 152-154): “*Following the in vitro validation, we measured the r1-r2* relaxivity in the living human brain. For this aim we calculated the linear dependency of R1 on R2* across voxels of different anatomically-defined ROIs*”. A detailed description of this process can be found in the method section “r1-r2* relaxivity computation for ROIs in the human brain”.

In order to approximate a voxel-wise relaxivity map, we developed an approach which is based on each voxel’s local neighborhood. This approach is detailed in Supplementary Section 3: Voxel-wise r1-r2* relaxivity visualization. We also show a comparison of the voxel-wise r1-r2* relaxivity to the R1 and R2* maps. Moreover, we provide an example for a voxel-wise visualization of the r1-r2* relaxivity approach in a representative meningioma patient, which demonstrates the Gd-free tumor enhancement. Importantly, for this patient we were able to replicate our ROI-based results on the voxel-wise level (Sup. Figure 12).

Finally, the reviewer’s comment was targeted at the biophysical model for the r1-r2* relaxivity. This model does not depend on the resolution of the measurements, and can therefore be applied to any estimation of the R1-R2* linear dependency. We therefore agree with the reviewer that in this context our phrasing was not clear. We changed this line in Supplementary Section 4.1 to clarify this point: “*The r1-r2* relaxivity measurement is defined as the linear dependency of R1 on R2* (within an ROI in the brain, across in vitro samples or across each voxel’s local neighborhood). This is equivalent to the total change in R1 values relative to the total change in R2* values*”.

10) P24: please state what are the units of Tf and Ft (units of concentration according to Eqs. S4 and S5).

We thank the reviewer for raising an interesting point. In the relaxivity model, the transferrin and ferritin concentrations are multiplied by their relaxivities. For example, R1 can be expressed by the following equation:

$$R_1 = r_{(1,Ft)}[Ft] + r_{(1,Tf)}[Tf] + r_{(1,M)}[M] .$$

The relaxivities ($r_{(1,Ft)}$, $r_{(1,Tf)}$, $r_{(1,M)}$) are measured in units of rate per concentration. The multiplication of relaxivities and concentrations (for example $r_{(1,Ft)}[Ft]$), produces measurements in units of rate (1/sec). Therefore, as long as $r_{(1,Ft)}$ and $[Ft]$ are measured in the same units of concentration, these units will cancel out. This means that the presented model for predicting relaxation rates does not depend on the units of concentration (as long as they are in agreement with the units of relaxivities). Specifically, when applying this model to in vitro and numerical simulations, we used units of [mg/ wet ml].

This is now described in the manuscript (Supplementary Section 4.2): *Where [Ft], [Tf] are the ferritin and transferrin concentrations respectively, measured in [mg/ wet ml]. [M] is the myelin concentration. $r_{(1,Ft)}$, $r_{(1,Tf)}$ and $r_{(1,M)}$ are the R1-relaxivities of ferritin, transferrin and myelin respectively. $r_{(2,Ft)}$, $r_{(2,Tf)}$ and $r_{(2,M)}$ are the R2*-relaxivities of ferritin, transferrin and myelin respectively. Notably, this model does not depend on the exact units of concentration as long as the relaxivities (rate per concentration) and concentrations measurements agree in units.*

11) Figure S19 caption describe four mixtures but the Figure shows six mixtures. Done.

12) Figure S21 is too small to see – please provide a zoomed image. Done.

13) P7L552: “For calibration” – please specific which calibration and also what acquisition scheme was used (apart from the preparation IR and post excitation spin-echo refocusing RF pulse).

We changed the phrasing to explain that the SEIR scans were used for B1+ mapping. A more detailed explanation can be found under the methods section “estimation of qMRI parameters for phantoms”:
Quantitative R1 & MTV mapping: R1 and MTV estimations for the lipid samples were computed with the mrQ⁴⁷ (<https://github.com/mezera/mrQ>) and Vista Lab (<https://github.com/vistalab/vistasoft/wiki>) software packages. The mrQ software was modified to suit the phantom system⁷⁹. The modification utilizes the fact that the Agar-Gd mixture which fills the box around the vials is homogeneous, and therefore can be assumed to have a constant R1 value. We used this gold-standard R1 value generated from the SEIR scans to correct for the excite bias in the SPGR scans”.

14) P28L588 & L594: Equations are missing numbers. Done.

15) P28L600-601 (and also relating to P22L405-407 and P22L414-416): MRI estimations of iron concentrations and relaxivities are done on a voxel level, producing an average value across each voxel. The iron concentrations calculated here, however, are based on content of ferritin and transferrin on a molecular length-scale. What is the interrelation between these two regimes? Can the molecular model apply to MRI-derived values which are averaged across relatively huge and heterogeneous MRI voxels?

We thank the reviewer for this comment. The referred equation relates iron-binding proteins concentrations to iron concentrations. We would like to clarify the constraints of this calculation. We agree with the reviewer that MRI estimations in the brain produce average values of the heterogeneous molecular environment of the tissue. We would like to stress that the iron concentration estimation mentioned by the reviewer, is only applicable to our in vitro experiments. These experiments are based on a controlled synthetic iron environment, which we assume to be homogeneous. Therefore, in these experiments the molecular makeup is relatively similar within and across all voxels. We do not apply this molecular model to MRI values of brain tissue which is complex and heterogenous. This is now clarified in the text (lines 634-636): *“Importantly, this model is only applicable in vitro, for controlled and homogeneous iron environments. We do not apply this molecular model to MRI values of brain tissue which is complex and heterogenous”.*

16) P29L630: please provide a reason for collecting MT weighted data.

We thank the reviewer for this comment. To establish that the r1-r2* relaxivity captures a novel brain contrast different from the dominant myelin-based MRI contrast, we compared the r1-r2* relaxivity to standard myelin-sensitive MRI parameters. In Sup. Figure 14 we show the correlations of R1, R2* and the r1-r2* relaxivity with MTsat. This analysis demonstrates that, unlike R1 and R2*, the r1-r2* relaxivity is not correlated with the myelin marker MTsat across the brain.

17) P31L675: please change B1 to B1+. Done.

18) P32L708 and elsewhere throughout the manuscript: please use subscripts for R1 and R2* where appropriate (i.e., R₁ and R₂*). Done.

19) P32L710-717 and P9L161-162: does this procedure of using a 5x5x5 moving window and relying on the local neighborhood of each voxel blur the ensuing r1-r2* maps?

We completely agree with the reviewer. The $r1-r2^*$ relaxivity provides a new MRI measurement sensitive to the iron homeostasis. However, this measurement is inherently limited in terms of resolution, due to the need to calculate the linear dependency of the parameters across voxels. This is true for both the ROI and sliding window voxel-wise implementations we used in the manuscript.

The limitations of the voxel-wise $r1-r2^*$ relaxivity approach are detailed in supplementary Section 3: Voxel-wise $r1-r2^*$ relaxivity visualization. Specifically, we discuss the blurring of the $r1-r2^*$ maps: *“Notably, the presented implementation still has some limitations. First, the moving-window approach used for calculating the local $r1-r2^*$ relaxivity leads to inherent smoothing. As a result, this approach is sensitive to partial volume effects for voxels on the border between tissue types, which could be driving the observed contrast between superficial and deep white matter and between tumor tissue and non-pathological tissue. In addition, the local computation of the $r1-r2^*$ relaxivity use fewer voxels compared to the ROI-based approach. It also does not include the binning procedure prior to the fitting which we used in the ROI-based approach. Therefore, this computation is less stable and is more sensitive to the inherent SNR of $R1$ and $R2^*$ ”*

20) P32L724: typo, remove the word “with”. Done.

21) P9L145-147: why does the stability of $r1-r2^*$ relaxivity across different myelin content (in vitro model) indicate that $r1-r2^*$ relaxivity has high specificity to differences in molecular state of the iron?

Our results indicate that different iron environments have distinct $r1-r2^*$ relaxivities even when varying the liposomal fractions (Sup. Figure 5f). This suggests that the $r1-r2^*$ relaxivity is specifically sensitive to the iron environment. While the lipid environment changes the magnitudes of $R1$ and $R2^*$, it does not affect their shared variation (the $r1-r2^*$ relaxivity). Yet, we agree with the reviewer that the referred sentence may be out of context in the main results section. We therefore changed this sentence (lines 149-150): *“These results suggest that the $r1-r2^*$ relaxivity is less sensitive to the lipid concentration and composition compared to $R1$ and $R2^*$.”*

22) Figure 2b: why does the left panel reflect the reliability of the method (as stated in the caption) rather than the variability of the measured values across different brain regions.

We thank the reviewer for this comment. We show that the variability across subjects in the $r1-r2^*$ relaxivity measurement within-region is smaller than the variability between brain regions. Therefore, this measurement is consistent across subjects and produces similar values, that highlights the significant and reliable contrast between brain regions. We changed the figure caption to clarify this: *“Fig. 2b: the $r1-r2^*$ relaxivity in different brain regions, the variation in each region is across normal subjects (age 27 ± 2 , $N = 21$). Within-region, this measurement is stable across subjects. It shows clear difference between regions, thus indicating its reliability.”*

23) P11L164-173 and Figure S13: WM regions are clearly separated using the newly suggested relaxivity biomarker. This is shown on a group-wise level in Figure S13. Does this also hold on a single subject level?

We now provide a new version of this figure, showing the results in the level of the single subject overlaid on the group averages. For most of the subjects, WM regions are clearly separated.

24) P12L206: $R2$ of 0.59 is reported while Figure 3c shows a value of 0.62.

We thank the reviewer for noticing this difference. We changed the reported value to 0.62.

25) Figure 4: shows R1 and R2* maps. Please also add a map of the r1-r2* relaxivity parameter of interest.

We thank the reviewer for this comment. We would like to highlight that the results in this figure are shown for the r1-r2* relaxivity measured in the ROI approach, where the linear fit is performed across voxels. A voxel-wise visualization of the r1-r2* relaxivity map in a meningioma patient, and a comparison of this contrast to R1 and R2*, can be found in Sup. Figure 11.

To clarify the difference between the ROI and the voxel-wise approaches we add a sentence to the figure legend: “*The r1-r2* relaxivity is calculated across voxels (R1 and R2* values shown in a) for each ROI. A voxel-wise visualization of the r1-r2* relaxivity available in Supplementary Section 3*”

26) Figure 4c-f: please mark the tumor ROI which was included in the analysis that is shown in these panels.

We thank the reviewer for this comment. We would like to clarify that these panels show the variability of the r1-r2* relaxivity measurements across 18 meningioma patients (for WM, GM and tumor tissues). Therefore, the presented results were not based on a single tumor ROI. An example of a tumor ROI is shown in panel a (the tumor is marked with an arrow). This point is now explained more clearly in the figure legend.

27) P18L278-279: $p > 0.01$ is considered as not statistically correlated. Conventionally, 0.05 is set as a statistically significant limit. Please add the specific p-value and elaborate why it is not considered significant.

We agree with the reviewer that this sentence was inaccurate. We changed this sentence and elaborated on the specific p-values (lines 285-287): “*This iron-related pathway was not significantly associated with R1 (FWER p -value > 0.05) and was less significantly associated with R2* (FWER p -value = 0.038)*”

The full table of p-values can be found in the supplementary (Sup. Table 1).

28) P19L314: commonly is à is commonly. Done.

29) P21L362: farther à further. Done.

30) Text that can use rephrasing to improve readability is marked with green underline in the attached PDF file. Done.

Reviewer #1 (Remarks to the Author):

The authors have comprehensively responded to my questions. I think this is a novel and important work that is ready for publication.

Scott Ayton

Reviewer #2 (Remarks to the Author):

See attached PDF document.

Reviewer #2 Attachment on the following page

Non-invasive assessment of normal and impaired iron homeostasis in 2023 living human brains

REVIEWER'S COMMENTS – REVISION 1

I thank the authors for thoroughly addressing all comments on the original manuscript version. Some minor comments remain and are listed below. Comments numbers are the same as in the original review cycle.

MAJOR COMMENTS

- 1) Satisfactorily addressed
 - I believe that Supp. Figure 36 has wrong x-label. It says “myelin (fraction)” which should be “myelin (concentration).
- 2) Thank you for adding the Table. Please note that:
 - a. Some of the parameters are missing units.
 - b. “reversed phase-encoding blips”. Is the meaning that the gradient directions were reversed? Or that the phase encoding direction was reversed? e.g., $A \gg P$ vs. $P \gg A$.
 - c. Please add the acquisition bandwidth to the list of parameters and whether any acceleration was used.

MINOR COMMENTS

- 3) Satisfactorily addressed.
- 4) Satisfactorily addressed.
- 5) Satisfactorily addressed.
- 6) Satisfactorily addressed.
- 7) Satisfactorily addressed.
- 8) Satisfactorily addressed.
- 9) Satisfactorily addressed.
- 10) Satisfactorily addressed.
- 11) Satisfactorily addressed.
- 12) Satisfactorily addressed.
- 13) Satisfactorily addressed.

I would suggest changing “excite bias” to “inhomogeneous excitation bias”.

- 14) Satisfactorily addressed.
- 15) Satisfactorily addressed.
- 16) Satisfactorily addressed.
- 17) Satisfactorily addressed.
- 18) Satisfactorily addressed.
- 19) Satisfactorily addressed.

Please consider moving this from Supp. Section 3 to the main manuscript.

- 20) Satisfactorily addressed.
- 21) Satisfactorily addressed.
- 22) Satisfactorily addressed.
- 23) Satisfactorily addressed.
- 24) Satisfactorily addressed.
- 25) Satisfactorily addressed.
- 26) Satisfactorily addressed.
- 27) I might be missing something. Isn't $p=0.038$ *more* statistically significant than $p>0.05$?
- 28) Satisfactorily addressed.
- 29) Satisfactorily addressed.
- 30) Satisfactorily addressed.

We thank the reviewer for the insightful comments on our manuscript throughout the revision process, we believe it contributed greatly to our work.

We have addressed the remaining minor comment in the final version of the manuscript (comments are in blue, our answers in black):

MAJOR COMMENTS

1) Satisfactorily addressed

- I believe that Supp. Figure 36 has wrong x-label. It says “myelin (fraction)” which should be “myelin (concentration).

In this analysis we simulate the fraction of myelin in the voxel and not the myelin concentration. We used MTV, an MRI myelin-marker, to estimate the myelin relaxivity in vivo. MTV is measured in units of fraction. Therefore, our simulations were based on fractional measurements of myelin.

2) Thank you for adding the Table. Please note that:

a. Some of the parameters are missing units.

We added units to all parameters.

b. “reversed phase-encoding blips”. Is the meaning that the gradient directions were reversed? Or that the phase encoding direction was reversed? e.g., $A \gg P$ vs. $P \gg A$.

The meaning is reversed phase encoding direction ($P \gg A$). We clarified this in the text.

c. Please add the acquisition bandwidth to the list of parameters and whether any acceleration was used.

Done

MINOR COMMENTS

3) Satisfactorily addressed.

4) Satisfactorily addressed.

5) Satisfactorily addressed.

6) Satisfactorily addressed.

7) Satisfactorily addressed.

8) Satisfactorily addressed.

9) Satisfactorily addressed.

10) Satisfactorily addressed.

11) Satisfactorily addressed.

12) Satisfactorily addressed.

13) Satisfactorily addressed.

I would suggest changing “excite bias” to “inhomogeneous excitation bias”.

Done

14) Satisfactorily addressed.

15) Satisfactorily addressed.

16) Satisfactorily addressed.

17) Satisfactorily addressed.

18) Satisfactorily addressed.

19) Satisfactorily addressed.

Please consider moving this from Supp. Section 3 to the main manuscript.

We now refer to this point in the discussion (lines 427-432): “Another limitation of our approach is that we estimate the dependency of R_1 on R_2^* across voxels, and it therefore produces one relaxivity measurement per anatomically-defined ROI. To visualize the contrast of the r_1 - r_2^*

relaxivity in the brain, we present a voxel-wise implementation based on a sliding-window approach. However, this implementation is more sensitive to partial volume and smoothing effects.”

20) Satisfactorily addressed.

21) Satisfactorily addressed.

22) Satisfactorily addressed.

23) Satisfactorily addressed.

24) Satisfactorily addressed.

25) Satisfactorily addressed.

26) Satisfactorily addressed.

27) I might be missing something. Isn't $p=0.038$ *more* statistically significant than $p>0.05$?

Our intention was that $p=0.038$ for the association with $R2^*$ is less statistically significant compared to the association with the $r1-r2^*$ relaxivity ($p<0.001$). We now clarify this in the text (lines 286-292): “The two most enriched pathways for the $r1-r2^*$ relaxivity were “immunoglobulin complex” (normalized enrichment score (NES)= -3.62, FWER p -value <0.001 ; Supplementary Fig. 41a) and “scavenging of heme from plasma” (NES= -3.27, FWER p -value <0.001 ; Supplementary Fig. 41b). While the former may relate to the response of the immune system to the cancerous process^{62,63}, the latter involves the absorption of free heme, a source of redox-active iron⁶⁴. This iron-related pathway was not significantly associated with $R1$ (FWER p -value >0.05) and was less significantly associated with $R2^*$ **compared to the $r1-r2^*$ relaxivity (FWER p -value=0.038).**”

28) Satisfactorily addressed.

29) Satisfactorily addressed.

30) Satisfactorily addressed.